# Effects of Graph Convolutions in Multi-layer Networks

**Aseem Baranwal**[*]  **Kimon Fountoulakis**[*]  **Aukosh Jagannath**[†]

## Abstract

Graph Convolutional Networks (GCNs) are one of the most popular architectures that are used to solve classification problems accompanied by graphical information. We present a rigorous theoretical understanding of the effects of graph convolutions in multi-layer networks. We study these effects through the node classification problem of a non-linearly separable Gaussian mixture model coupled with a stochastic block model. First, we show that a single graph convolution expands the regime of the distance between the means where multi-layer networks can classify the data by a factor of at least $1/\sqrt[4]{\deg}$, where $\deg$ denotes the expected degree of a node. Second, we show that with a slightly stronger graph density, two graph convolutions improve this factor to at least $1/\sqrt[4]{n}$, where $n$ is the number of nodes in the graph. Finally, we provide both theoretical and empirical insights into the performance of graph convolutions placed in different combinations among the layers of a neural network, concluding that the performance is mutually similar for all combinations of the placement. We present extensive experiments on both synthetic and real-world data that illustrate our results.

## 1 Introduction

A large amount of interesting data and the practical challenges associated with them are defined in the setting where entities have attributes as well as information about mutual relationships. Traditional classification models have been extended to capture such relational information through graphs (Hamilton, 2020), where each node has individual attributes and the edges of the graph capture the relationships among the nodes. A variety of applications characterized by this type of graph-structured data include works in the areas of social analysis (Backstrom & Leskovec, 2011), recommendation systems (Ying et al., 2018), computer vision (Monti et al., 2017), study of the properties of chemical compounds (Gilmer et al., 2017; Scarselli et al., 2009), statistical physics (Bapst et al., 2020; Battaglia et al., 2016), and financial forensics (Zhang et al., 2017; Weber et al., 2019).

The most popular learning models for relational data use graph convolutions (Kipf & Welling, 2017), where the idea is to aggregate the attributes of the set of neighbours of a node instead of only utilizing its own attributes. Despite several empirical studies of various GCN-type models (Chen et al., 2019; Ma et al., 2022) that demonstrate an improvement in the performance of traditional classification methods such as MLPs, there has been limited progress in the theoretical understanding of the benefits of graph convolutions in multi-layer networks in terms of improving node classification tasks.

**Related work.** The capacity of a graph convolution for one-layer networks is studied in Baranwal et al. (2021), along with its out-of-distribution (OoD) generalization potential. A more recent work (Wu et al., 2022) formulates the node-level OoD problem, and develops a learning method that facilitates GNNs to leverage invariance principles for prediction. In Gasteiger et al. (2019), the authors utilize a propagation scheme based on personalized PageRank to construct a model that outperforms several GCN-like methods for semi-supervised classification. Through their algorithm, APPNP, they show that placing power iterations at the last layer of an MLP achieves state of the art performance. Our results align with this observation.

---

[*]David R. Cheriton School of Computer Science, University of Waterloo, Waterloo, Canada. `{aseem.baranwal,kimon.fountoulakis}@uwaterloo.ca`

[†]Department of Statistics and Actuarial Science, Department of Applied Mathematics, University of Waterloo, Waterloo, Canada. `a.jagannath@uwaterloo.ca`

There exists a large amount of theoretical work on unsupervised learning for random graph models where node features are absent and only relational information is available (Decelle et al., 2011; Massoulié, 2014; Mossel et al., 2018; 2015; Abbe & Sandon, 2015; Abbe et al., 2015; Bordenave et al., 2015; Deshpande et al., 2015; Montanari & Sen, 2016; Banks et al., 2016; Abbe & Sandon, 2018; Li et al., 2019; Kloumann et al., 2017; Gaudio et al., 2022). For a comprehensive survey, see Abbe (2018); Moore (2017). For data models which have node features coupled with relational information, several works have studied the semi-supervised node classification problem, see, for example, Scarselli et al. (2009); Cheng et al. (2011); Gilbert et al. (2012); Dang & Viennet (2012); Günnemann et al. (2013); Yang et al. (2013); Hamilton et al. (2017); Jin et al. (2019); Mehta et al. (2019); Chien et al. (2022); Yan et al. (2021). These papers provide good empirical insights into the merits of graph structure in the data. We complement these studies with theoretical results that explain the effects of graph convolutions in a multi-layer network.

In Deshpande et al. (2018); Lu & Sen (2020), the authors explore the fundamental thresholds for the classification of a substantial fraction of the nodes with linear sample complexity and large but finite degree. Another relatively recent work (Hou et al., 2020) proposes two graph smoothness metrics for measuring the benefits of graphical information, along with a new attention-based framework. In Fountoulakis et al. (2022), the authors provide a theoretical study of the graph attention mechanism (GAT) and identify the regimes where the attention mechanism is (or is not) beneficial to node-classification tasks. Our study focuses on convolutions instead of attention-based mechanisms. Several works have studied the expressive power and extrapolation of GNNs, along with the oversmoothing phenomenon (see, for e.g., Balcilar et al. (2021); Xu et al. (2021); Oono & Suzuki (2020); Li et al. (2018)). Some other works have also studied the homophily and heterophily problem in GNNs (Luan et al., 2021; Ma et al., 2022). However, our focus is to draw a comparison of the benefits and limitations of graph convolutions with those of a traditional MLP that does not utilize relational information. Similar to Wei et al. (2022), our setting is immune to the heterophily problem, and the focus of our study is on regimes where oversmoothing does not occur.

To the best of our knowledge, this area of research still lacks theoretical guarantees that explain when and why graphical data, and in particular, graph convolutions, can boost traditional multi-layer networks to perform better on node-classification tasks. To this end, we study the effects of graph convolutions in deeper layers of a multi-layer network. For node classification tasks, we also study whether one can avoid using additional layers in the network design for the sole purpose of gathering information from neighbours that are farther away, by comparing the benefits of placing all convolutions in a single layer versus placing them in different layers.

**Our contributions.** We study the performance of multi-layer networks for the task of binary node classification on a data model where node features are sampled from a Gaussian mixture, and relational information is sampled from a symmetric two-block stochastic block model[1] (see Section 2.1 for details). The node features are modelled after XOR data with two classes, and therefore, has four distinct components, two for each class. Our choice of the data model is inspired from the fact that it is non-linearly separable. Hence, a single layer network fails to classify the data from this model. Similar data models based on the contextual stochastic block model (CSBM) have been used extensively in the literature, see, for example, Deshpande et al. (2018); Binkiewicz et al. (2017); Chien et al. (2021; 2022); Baranwal et al. (2021). We now summarize our contributions below.

1. We show that when node features are accompanied by a graph, a single graph convolution enables a multi-layer network to classify the nodes in a wider regime as compared to methods that do not utilize the graph, improving the threshold for the distance between the means of the features by a factor of at least $1/\sqrt[4]{\mathbb{E}\deg}$. Furthermore, assuming a slightly denser graph, we show that with two graph convolutions, a multi-layer network can classify the data in an even wider regime, improving the threshold by a factor of at least $1/\sqrt[4]{n}$, where $n$ is the number of nodes in the graph.

2. We show that for multi-layer networks equipped with graph convolutions, the classification capacity is determined by the number of graph convolutions rather than the number of layers in the network. In particular, we study the gains obtained by placing graph convolutions in a layer, and compare the benefits of placing all convolutions in a single layer versus

---

[1] Our analyses generalize to non-symmetric SBMs with more than two blocks. However, we focus on the binary symmetric case for the sake of simplicity in the presentation of our ideas.

placing them in different combinations across different layers. We find that the performance is mutually similar for all combinations with the same number of graph convolutions.

3. We verify our theoretical results through extensive experiments on both synthetic and real-world data, showing trends about the performance of graph convolutions in various combinations across multiple layers of a network, and in different regimes of interest.

The rest of our paper is organized as follows: In Section 2, we provide a detailed description of the data model and the network architecture that is central to our study, followed by our analytical results in Section 3. Finally, Section 4 presents extensive experiments that illustrate our theoretical findings.

## 2 PRELIMINARIES

### 2.1 DESCRIPTION OF THE DATA MODEL

Let $n, d$ be positive integers, where $n$ denotes the number of data points (sample size) and $d$ denotes the dimension of the features. Define the Bernoulli random variables $\varepsilon_1, \ldots, \varepsilon_n \sim \mathrm{Ber}(1/2)$ and $\eta_1, \ldots, \eta_n \sim \mathrm{Ber}(1/2)$. Further, define two classes $C_b = \{i \in [n] \mid \varepsilon_i = b\}$ for $b \in \{0, 1\}$.

Let $\boldsymbol{\mu}$ and $\boldsymbol{\nu}$ be fixed vectors in $\mathbb{R}^d$, such that $\|\boldsymbol{\mu}\|_2 = \|\boldsymbol{\nu}\|_2$ and $\langle \boldsymbol{\mu}, \boldsymbol{\nu} \rangle = 0$.[2] Denote by $\mathbf{X} \in \mathbb{R}^{n \times d}$ the data matrix where each row-vector $\mathbf{X}_i \in \mathbb{R}^d$ is an independent Gaussian random vector distributed as $\mathbf{X}_i \sim \mathcal{N}((2\eta_i - 1)((1 - \varepsilon_i)\boldsymbol{\mu} + \varepsilon_i\boldsymbol{\nu}), \sigma^2)$. We use the notation $\mathbf{X} \sim \text{XOR-GMM}(n, d, \boldsymbol{\mu}, \boldsymbol{\nu}, \sigma^2)$ to refer to data sampled from this model.

Let us now define the model with graphical information. In this case, in addition to the features $\mathbf{X}$ described above, we have a graph with the adjacency matrix, $\mathbf{A} = (a_{ij})_{i,j \in [n]}$, that corresponds to an undirected graph including self-loops, and is sampled from a standard symmetric two-block stochastic block model with parameters $p$ and $q$, where $p$ is the intra-block and $q$ is the inter-block edge probability. The $\text{SBM}(n, p, q)$ is then coupled with the XOR-GMM$(n, d, \boldsymbol{\mu}, \boldsymbol{\nu}, \sigma^2)$ in the way that $a_{ij} \sim \mathrm{Ber}(p)$ if $\varepsilon_i = \varepsilon_j$ and $a_{ij} \sim \mathrm{Ber}(q)$ if $\varepsilon_i \neq \varepsilon_j$. For data $(\mathbf{A}, \mathbf{X}) = (\{a_{ij}\}_{i,j \in [n]}, \{\mathbf{X}_i\}_{i \in [n]})$ sampled from this model, we say $(\mathbf{A}, \mathbf{X}) \sim \text{XOR-CSBM}(n, d, \boldsymbol{\mu}, \boldsymbol{\nu}, \sigma^2, p, q)$.

We will denote by $\mathbf{D}$ the diagonal degree matrix of the graph with adjacency matrix $\mathbf{A}$, and thus, $\mathbf{deg}(i) = \mathbf{D}_{ii} = \sum_{j=1}^{n} a_{ij}$ denotes the degree of node $i$. We will use $N_i = \{j \in [n] \mid a_{ij} = 1\}$ to denote the set of neighbours of a node $i$. We will also use the notation $i \sim j$ or $i \nsim j$ throughout the paper to signify, respectively, that $i$ and $j$ are in the same class, or in different classes.

### 2.2 NETWORK ARCHITECTURE

Our analysis focuses on MLP architectures with ReLU activations. In particular, for a network with $L$ layers, we define the following:

$$\mathbf{H}^{(0)} = \mathbf{X},$$
$$\left.\begin{aligned} f^{(l)}(\mathbf{X}) &= (\mathbf{D}^{-1}\mathbf{A})^{k_l}\mathbf{H}^{(l-1)}\mathbf{W}^{(l)} + \mathbf{b}^{(l)} \\ \mathbf{H}^{(l)} &= \mathrm{ReLU}(f^{(l)}(\mathbf{X})) \end{aligned}\right\} \text{ for } l \in [L],$$
$$\hat{\mathbf{y}} = \varphi(f^{(L)}(\mathbf{X})).$$

Here, $\mathbf{X} \in \mathbb{R}^{n \times d}$ is the given data, which is an input for the first layer and $\varphi(x) = \mathrm{sigmoid}(x) = \frac{1}{1 + e^{-x}}$, applied element-wise. The final output of the network is represented by $\hat{\mathbf{y}} = \{\hat{y}_i\}_{i \in [n]}$. Note that $\mathbf{D}^{-1}\mathbf{A}$ is the normalized adjacency matrix[3] and $k_l$ denotes the number of graph convolutions placed in layer $l$. In particular, for a simple MLP with no graphical information, we have $\mathbf{A} = \mathbf{I}_n$.

We will denote by $\theta$, the set of all weights and biases, $(\mathbf{W}^{(l)}, \mathbf{b}^{(l)})_{l \in [L]}$, which are the learnable parameters of the network. For a dataset $(\mathbf{X}, \mathbf{y})$, we denote the binary cross-entropy loss obtained by

---

[2]We take $\boldsymbol{\mu}$ and $\boldsymbol{\nu}$ to be orthogonal and of the same magnitude for keeping the calculations relatively simpler, while clearly depicting the main ideas behind our results.

[3]Our results rely on degree concentration for each node, hence, they readily generalize to other normalization methods like $\mathbf{D}^{-\frac{1}{2}}\mathbf{A}\mathbf{D}^{-\frac{1}{2}}$ (see Appendix A for proofs).

a multi-layer network with parameters $\theta$ by $\ell_\theta(\mathbf{A}, \mathbf{X}) = -\frac{1}{n} \sum_{i \in [n]} y_i \log(\hat{y}_i) + (1 - y_i) \log(1 - \hat{y}_i)$, and the optimization problem is formulated as

$$\text{OPT}(\mathbf{A}, \mathbf{X}) = \min_{\theta \in \mathcal{C}} \ell_\theta(\mathbf{A}, \mathbf{X}), \tag{1}$$

where $\mathcal{C}$ denotes a suitable constraint set for $\theta$. For our analyses, we take the constraint set $\mathcal{C}$ to impose the condition $\left\| \mathbf{W}^{(1)} \right\|_2 \leq R$ and $\left\| \mathbf{W}^{(l)} \right\|_2 \leq 1$ for all $1 < l \leq L$, i.e., the weight parameters of all layers $l > 1$ are normalized, while for $l = 1$, the norm is bounded by some fixed value $R$. This is necessary because without the constraint, the value of the loss function can go arbitrarily close to $0$. Furthermore, the parameter $R$ helps us concisely provide bounds for the loss in our theorems for various regimes by bounding the Lipschitz constant of the learned function. In the rest of our paper, we use $\ell_\theta(\mathbf{X})$ to denote $\ell_\theta(\mathbf{I}_n, \mathbf{X})$, which is the loss in the absence of graphical information.

## 3  RESULTS

We now describe our theoretical contributions, followed by a discussion and a proof sketch.

### 3.1  SETTING UP THE BASELINE

Before stating our main result about the benefits and performance of graph convolutions, we set up a comparative baseline in the setting where graphical information is absent. In the following theorem, we completely characterize the classification threshold for the XOR-GMM data model in terms of the distance between the means of the mixture model and the number of data points $n$. Let $\Phi(\cdot)$ denote the cumulative distribution function of a standard Gaussian, and $\Phi_c(\cdot) = 1 - \Phi(\cdot)$.

**Theorem 1.** *Let $\mathbf{X} \in \mathbb{R}^{n \times d} \sim$ XOR-GMM$(n, d, \boldsymbol{\mu}, \boldsymbol{\nu}, \sigma^2)$ and define $\gamma = \|\boldsymbol{\mu} - \boldsymbol{\nu}\|_2$ to be the distance between the means. Then we have the following:*

1. *Assume that $\gamma \leq K\sigma$ and let $h(\mathbf{x}) : \mathbb{R}^d \to \{0, 1\}$ be any binary classifier. Then for any $K > 0$ and any $\epsilon \in (0, 1)$, at least a fraction $2\Phi_c \left( K/2 \right)^2 - O(n^{-\epsilon/2})$ of all data points are misclassified by $h$ with probability at least $1 - \exp(-2n^{1-\epsilon})$.*

2. *For any $\epsilon > 0$, if the distance between the means is $\gamma = \Omega(\sigma(\log n)^{\frac{1}{2} + \epsilon})$, then for any $c > 0$, with probability at least $1 - O(n^{-c})$, there exist a two-layer and a three-layer network that perfectly classify the data, and obtain a cross-entropy loss given by*

$$\ell_\theta(\mathbf{X}) = C \exp\left( -\frac{R}{\sqrt{2}} \gamma \left( 1 \pm \sqrt{c}/(\log n)^\epsilon \right) \right),$$

*where $C \in [1/2, 1]$ is an absolute constant and $R$ is the optimality constraint from Eq. (1).*

Part one of Theorem 1 shows that if the means of the features of the two classes are at most $O(\sigma)$ apart then with overwhelming probability, there is a constant fraction of points that are misclassified. Note that the fraction of misclassified points is $2\Phi_c(K/2)^2$, which approaches $0$ as $K \to \infty$ and approaches $1/2$ as $K \to 0$, signifying that if the means are very far apart then we successfully classify all data points, while if they coincide then we always misclassify roughly half of all data points. Furthermore, note that if $K = c\sqrt{\log n}$ for some constant $c \in [0, 1)$, then the total number of points misclassified is $2n\Phi_c(K)^2 \asymp \frac{n}{K^2} e^{-K^2} \asymp \frac{n^{1-c^2}}{\log n} = \Omega(1)$. Thus, intuitively, $K \asymp \sqrt{\log n}$ is the threshold beyond which learning methods are expected to perfectly classify the data. This is formalized in part two of the theorem, which supplements the misclassification result by showing that if the means are roughly $\omega(\sigma\sqrt{\log n})$ apart then the data is classifiable with overwhelming probability.

### 3.2  IMPROVEMENT THROUGH GRAPH CONVOLUTIONS

We now state the results that explain the effects of graph convolutions in multi-layer networks with the architecture described in Section 2.2. We characterize the improvement in the classification threshold in terms of the distance between the means of the node features. Let $\text{erf}(t) = 2\Phi(t\sqrt{2}) - 1$ be the Gauss error function and $\zeta(t) = t \, \text{erf}(t) - (1 - \exp(-t^2))/\sqrt{\pi}$.

**Theorem 2.** *Let $(\mathbf{A}, \mathbf{X}) \sim XOR\text{-}CSBM(n, d, \boldsymbol{\mu}, \boldsymbol{\nu}, \sigma^2, p, q)$, $\gamma = \|\boldsymbol{\mu} - \boldsymbol{\nu}\|_2$, and $\Gamma(p, q) = |p - q|/(p + q)$. There exist a two-layer network and a three-layer network with the following properties:*

- *If the intra-class and inter-class edge probabilities are $p, q = \Omega(\frac{\log^2 n}{n})$, and it holds that $\Gamma(p, q)\zeta(\gamma/2\sigma) = \omega\left(\sqrt{\frac{\log n}{n(p+q)}}\right)$, then for any $c > 0$, with probability at least $1 - O(n^{-c})$, the networks equipped with a graph convolution in the second or the third layer perfectly classify the data, and obtain the following loss:*

$$\ell_\theta(\mathbf{A}, \mathbf{X}) = C' \exp\left(-C\sigma R\Gamma(p, q)\zeta(\gamma/2\sigma)(1 \pm \sqrt{c/\log n})\right),$$

  *where $C > 0$ and $C' \in [1/2, 1]$ are constants and $R$ is the constraint from Eq. (1).*

- *If $p, q = \Omega(\frac{\log n}{\sqrt{n}})$ and $\Gamma(p, q)^2 \zeta(\gamma/2\sigma) = \omega\left(\sqrt{\frac{\log n}{n}}\right)$, then for any $c > 0$, with probability at least $1 - O(n^{-c})$, the networks with any combination of two graph convolutions in the second and/or the third layers perfectly classify the data, and obtain the following loss:*

$$\ell_\theta(\mathbf{A}, \mathbf{X}) = C' \exp\left(-C\sigma R\Gamma(p, q)^2 \zeta(\gamma/\sigma)(1 \pm \sqrt{c/\log n})\right),$$

  *where $C > 0$ and $C' \in [1/2, 1]$ are constants and $R$ is the constraint from Eq. (1).*

To understand Theorem 2, it helps to consider the regime where $\Gamma(p, q) = \Omega(1)$. Part one of the theorem shows that under the assumption that $p, q = \Omega(\log^2 n/n)$, a single graph convolution improves the classification threshold for $\gamma$, the distance between the means by a factor of at least $1/\sqrt[4]{n(p+q)}$ as compared to the case without the graph. Part two then shows that with a slightly stronger assumption on the graph density, we observe further improvement in the threshold up to a factor of at least $1/\sqrt[4]{n}$. We refer to Appendix A.8 for a comprehensive explanation of this simpler case.

Note that although the regime of graph density is different for part two of the theorem, the result itself is an improvement. In particular, if $p, q = \Omega(\log n/\sqrt{n})$ then part one of the theorem states that one graph convolution achieves an improvement of at least $1/\sqrt[8]{n}$, while part two states that two convolutions improve it to at least $1/\sqrt[4]{n}$. However, we also emphasize that in the regime where the graph is dense, i.e., when $p, q = \Omega_n(1)$, two graph convolutions do not have a significant advantage over one graph convolution. Our experiments in Section 4.1 demonstrate this effect.

The XOR-CSBM data model also demonstrates why graph convolutions in the first layer can severely hurt the classification accuracy. Hence, for Theorem 2, our analysis only considers networks with no graph convolution in the first layer, i.e., $k_1 = 0$. This effect is visualized in Fig. 1, and is attributed to the averaging of data points in the same class but different components of the mixture that have means with opposite signs. We defer the reader to Appendix A.5 for a more formal argument, and to Appendix B.1 for experiments that demonstrate this phenomenon. As $n$ (the sample size) grows, the difference between the averages of node features over the two classes diminishes (see Figs. 1a and 1b). In other words, the means of the two classes collapse to the same point for large $n$. However, in the last layer, since the input consists of transformed features that are linearly separable, a graph convolution helps with the classification task (see Figs. 1c and 1d).

### 3.3 PLACEMENT OF GRAPH CONVOLUTIONS

We observe that the improvements in the classification capability of a multi-layer network depends on the number of convolutions, and does not depend on where the convolutions are placed. In particular, for the XOR-CSBM data model, putting the same number of convolutions among the second and/or the third layer in any combination achieves mutually similar improvements in the classification task.

**Corollary 2.1.** *Consider the data model $XOR\text{-}CSBM(n, d, \boldsymbol{\mu}, \boldsymbol{\nu}, \sigma^2, p, q)$ and the network architecture from Section 2.2.*

- *Assume that $p, q = \Omega(\log^2 n/n)$, and consider the three-layer network characterized by part one of Theorem 2, with one graph convolution. For this network, placing the graph convolution in the second layer ($k_2 = 1, k_3 = 0$) obtains the same results as placing it in the third layer ($k_2 = 0, k_3 = 1$).*

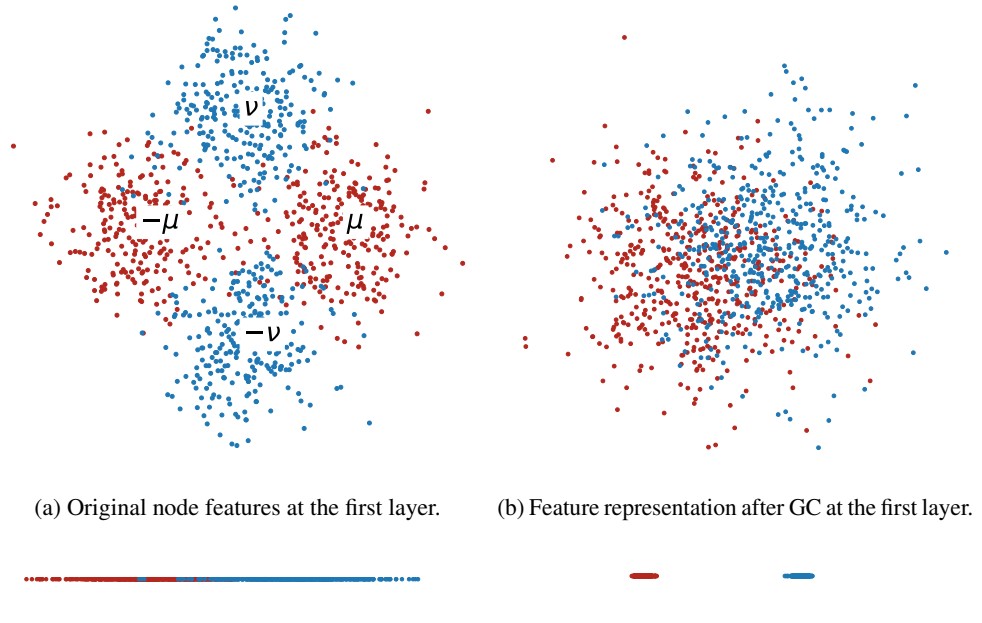

(a) Original node features at the first layer.  (b) Feature representation after GC at the first layer.

(c) Feature representation at the last layer.  (d) Feature representation after GC at the last layer.

Figure 1: Placement of a graph convolution (GC) in the first layer versus the last layer for data sampled from the XOR-CSBM. For this figure we used $1000$ nodes in each class and a randomly sampled stochastic block-model graph with $p = 0.8$ and $q = 0.2$.

- *Assume that $p, q = \Omega(\log n / \sqrt{n})$, and consider the three-layer network characterized by part two of Theorem 2, with two graph convolutions. For this network, placing both convolutions in the second layer ($k_2 = 2, k_3 = 0$) or both of them in the third layer ($k_2 = 0, k_3 = 2$) obtains the same results as placing one convolution in the second layer and one in the third layer ($k_2 = 1, k_3 = 1$).*

Corollary 2.1 is immediate from the proof of Theorem 2 (see Appendices A.6 and A.7). In Section 4, we also show extensive experiments on both synthetic and real-world data that demonstrate this result.

### 3.4 PROOF SKETCH

In this section, we provide an overview of the key ideas and intuition behind our proof technique for the results. For comprehensive proofs, see Appendix A.

For part one of Theorem 1, we utilize the assumption on the distribution of the data. Since the underlying distribution of the mixture model is known, we can find the (Bayes) optimal classifier[4], $h^*(\mathbf{x})$, for the XOR-GMM, which takes the form $h^*(\mathbf{x}) = \mathbb{1}(|\langle \mathbf{x}, \boldsymbol{\nu} \rangle| - |\langle \mathbf{x}, \boldsymbol{\mu} \rangle|)$, where $\mathbb{1}(\cdot)$ is the indicator function. We then compute a lower bound on the probability that $h^*$ fails to classify one data point from this model, followed by a concentration argument that computes a lower bound on the fraction of points that $h^*$ fails to classify with overwhelming probability. Consequently, a negative result for the Bayes optimal classifier implies a negative result for all classifiers.

For part two of Theorem 1, we design a two-layer and a three-layer network that realize the (Bayes) optimal classifier. We then use a concentration argument to show that in the regime where the distance between the means is large enough, the function representing our two-layer or three-layer network roughly evaluates to a quantity that has a positive sign for one class and a negative sign for the other class. Furthermore, the output of the function scales with the distance between the means. Thus, with a suitable assumption on the magnitude of the distance between the means, the output of the

---

[4]A Bayes classifier makes the most probable prediction for a data point. Formally, such a classifier is of the form $h^*(\boldsymbol{x}) = \operatorname{argmax}_{b \in \{0,1\}} \mathbf{Pr}[y = b \mid \boldsymbol{x}]$.

networks has the correct signs with overwhelming probability. Following this argument, we show that the cross-entropy loss obtained by the networks can be made arbitrarily small by controlling the optimization constraint $R$ (see Eq. (1)), implying perfect classification.

For Theorem 2, we observe that for the (Bayes) optimal networks designed for Theorem 1, placing graph convolutions in the second or the third layer reduces the effective variance of the functions representing the network. This stems from the fact that for the data model we consider, multi-layer networks with $\mathrm{ReLU}$ activations are Lipschitz functions of Gaussian random variables. First, we compute the precise reduction in the variance of the data characterized by $K > 0$ graph convolutions (see Lemma A.3). Then for part one of the theorem where we analyze one graph convolution, we use the assumption on the graph density to conclude that the degrees of each node concentrate around the expected degree. This helps us characterize the variance reduction, which further allows the distance between the means to be smaller than in the case of a standard MLP, hence, obtaining an improvement in the threshold for perfect classification[5]. Part two of the theorem studies the placement of two graph convolutions using a very similar argument. In this case, the variance reduction is characterized by the number of common neighbours of a pair of nodes rather than the degree of a node, and is stronger than the variance reduction offered by a single graph convolution.

## 4  EXPERIMENTS

In this section we provide empirical evidence that supports our claims in Section 3. We begin by analyzing the synthetic data models XOR-GMM and XOR-CSBM that are crucial to our theoretical results, followed by a similar analysis on multiple real-world datasets tailored for node classification tasks. We show a comparison of the test accuracy obtained by various learning methods in different regimes, along with a display of how the performance changes with the properties of the underlying graph, i.e., with the intra-class and inter-class edge probabilities $p$ and $q$.

For both synthetic and real-world data, the performance of the networks does not change significantly with the choice of the placement of graph convolutions. In particular, placing all convolutions in the last layer achieves a similar performance as any other placement for the same number of convolutions. This observation aligns with the results in Gasteiger et al. (2019).

### 4.1  SYNTHETIC DATA

In this section, we empirically show the landscape of the accuracy achieved for various multi-layer networks with up to three layers and up to two graph convolutions[6]. In Fig. 2, we show that as claimed in Theorem 2, a single graph convolution reduces the classification threshold by a factor of $1/\sqrt[4]{\mathbb{E}\,\mathbf{deg}}$ and two graph convolutions reduce the threshold by a factor of $1/\sqrt[4]{n}$, where $\mathbb{E}\,\mathbf{deg} = \frac{n}{2}(p + q)$.

We observe that the placement of graph convolutions does not matter as long as it is not in the first layer. Figs. 2a and 2b show that the performance is mutually similar for all networks that have one graph convolution placed in the second or the third layer, and for all networks that have two graph convolutions placed in any combination among the second and the third layers. In Figs. 2c and 2d, we observe that two graph convolutions do not obtain a significant advantage over one graph convolution in the setting where $p$ and $q$ are large, i.e., when the graph is dense. We observed similar results for various other values of $p$ and $q$ (see Appendix B.1 for some more plots).

Furthermore, in Appendix B.1 we verify that if a graph convolution is placed in the first layer of a network, then it is difficult to learn a classifier for the XOR-CSBM data model. In this case, test accuracy is low even for the regime where the distance between the means is quite large.

### 4.2  REAL-WORLD DATA

For real-world data, we test our results on three graph benchmarks: *CORA*, *CiteSeer*, and *Pubmed* citation network datasets (Sen et al., 2008). Results for larger datasets are presented in Appendix B.2. We observe the following trends: First, as claimed in Theorem 2, networks that utilize the graph

---

[5]We note that although scaling the parameters of a network scales the output of the network, yet, it does not affect the accuracy, which is determined by the sign of the outputs.

[6]We defer to Appendix B.1 for experiments with networks having graph convolutions in the first layer.

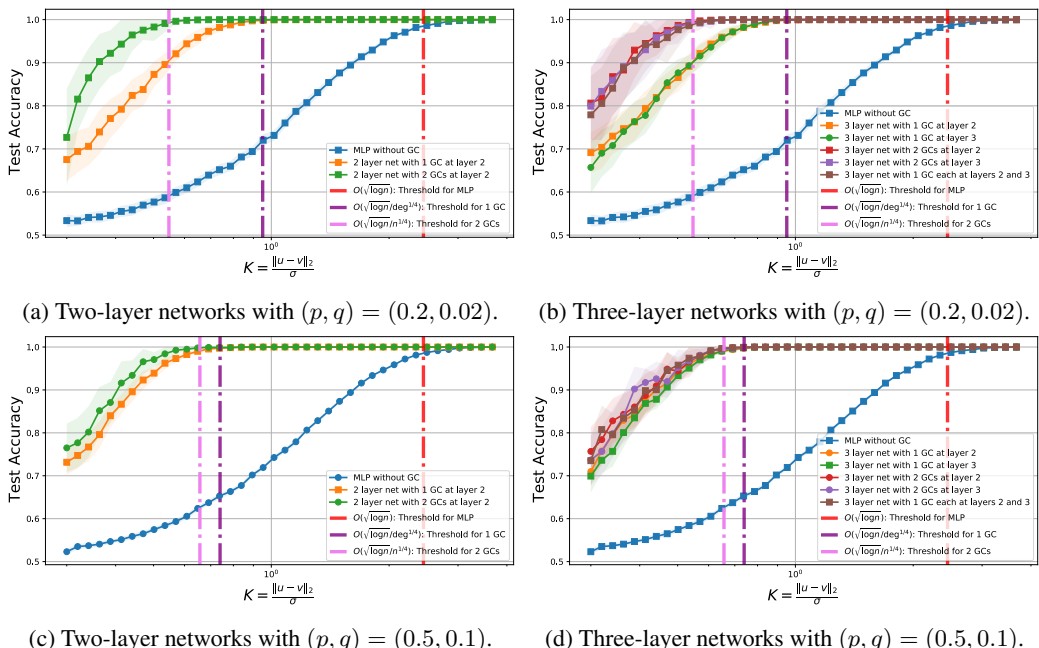

(a) Two-layer networks with $(p, q) = (0.2, 0.02)$.

(b) Three-layer networks with $(p, q) = (0.2, 0.02)$.

(c) Two-layer networks with $(p, q) = (0.5, 0.1)$.

(d) Three-layer networks with $(p, q) = (0.5, 0.1)$.

Figure 2: Averaged test accuracy (over $50$ trials) for various networks with and without graph convolutions on the XOR-CSBM data model with $n = 400, d = 4$ and $\sigma^2 = 1/d$. The x-axis denotes the ratio $K = \|\boldsymbol{\mu} - \boldsymbol{\nu}\|_2 / \sigma$ on a logarithmic scale. The vertical lines indicate the classification thresholds mentioned in part two of Theorem 1 (red), and in Theorem 2 (violet and pink).

perform remarkably better than a traditional MLP that does not use relational information. Second, all networks with one graph convolution in any layer achieve a mutually similar performance, and all networks with two graph convolutions in any combination of placement achieve a mutually similar performance. This demonstrates a result similar to Corollary 2.1 for real-world data. Finally, networks with two graph convolutions perform better than networks with one graph convolution.

In Fig. 3, we present for all networks, the maximum accuracy over $50$ trials, where each trial corresponds to a random initialization of the networks. For 2-layer networks, the hidden layer has width $16$, and for 3-layer networks, both hidden layers have width $16$. We use a dropout probability of $0.5$ and a weight decay of $10^{-5}$ while training.

For this study, we attribute minor changes in the accuracy to hyperparameters involving dropout and weight decay. This helps us clearly observe the important difference in the accuracy of networks with one graph convolution versus two graph convolutions. For example, in Fig. 3a, we note that there are differences in the accuracy of the networks with one graph convolution (red and blue). However, these differences are minor compared to the networks with one convolution (red and blue) and networks with two convolutions (green and yellow). We also show the averaged accuracy in Appendix B.2. Note that the accuracy slightly differs from well-known results in the literature due to implementation differences. In particular, the GCN implementation in Kipf & Welling (2017) uses $\tilde{\mathbf{A}} = \mathbf{D}^{-\frac{1}{2}}\mathbf{A}\mathbf{D}^{-\frac{1}{2}}$ as the normalized adjacency matrix, however, we use $\tilde{\mathbf{A}} = \mathbf{D}^{-1}\mathbf{A}$.[7] In Appendix B.2, we also show empirical results for the normalization $\tilde{\mathbf{A}} = \mathbf{D}^{-\frac{1}{2}}\mathbf{A}\mathbf{D}^{-\frac{1}{2}}$.

## 5  CONCLUSION AND FUTURE WORK

We study the fundamental limits of the capacity of graph convolutions when placed beyond the first layer of a multi-layer network for the XOR-CSBM data model, and provide theoretical guarantees for their performance in different regimes of the signal in the data. Through our experiments on both synthetic and real-world data, we show that the number of convolutions is a more significant

---

[7]Our proofs rely on degree concentration, and thus, generalize to the other type of normalization as well.

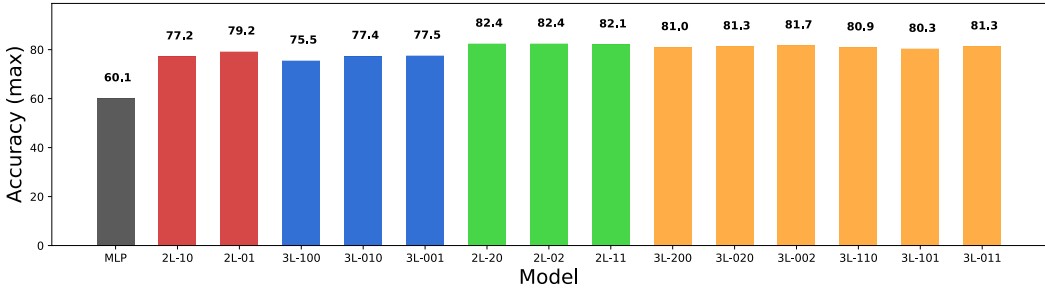

(a) Accuracy of various learning models on the CORA dataset.

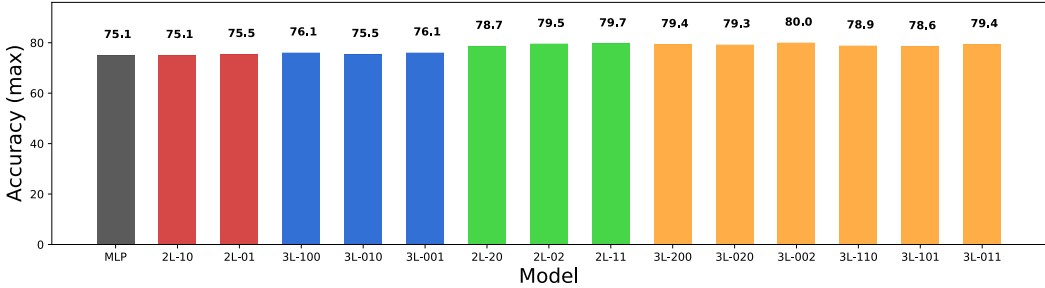

(b) Accuracy of various learning models on the Pubmed dataset.

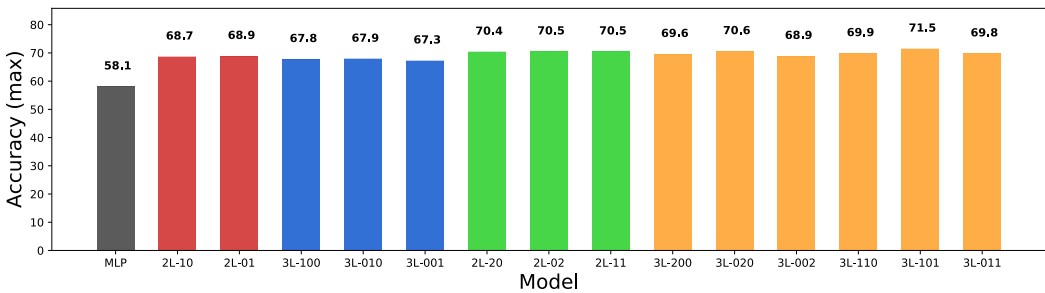

(c) Accuracy of various learning models on the CiteSeer dataset.

Figure 3: Maximum accuracy (percentage) over 50 trials for various networks. A network with $k$ layers and $j_1, \ldots, j_k$ convolutions in each of the layers is represented by the label $k$L-$j_1 \ldots j_k$.

factor for determining the performance of a network, rather than the number of layers in the network. Furthermore, we show that placing graph convolutions in any combination achieves mutually similar performance enhancements for the same number of them. We observe that multiple graph convolutions are advantageous when the underlying graph is relatively sparse. Intuitively, this is because in a dense graph, a single convolution can gather information from a large number of nodes, while in a sparser graph, more convolutions are needed to gather information from a larger number of nodes.

Our analysis is limited to a positive result and we only provide a minimum guarantee for improvement in the classification threshold. To fully understand the limitations of graph convolutions, a complementary negative result (similar to part one of Theorem 1) for data models with relational information is required, showing the maximum improvement that graph convolutions can realize in a multi-layer network. This problem is hard because a graph convolution transforms an iid set of features into a highly correlated set of features, making it difficult to apply classical high-dimensional concentration arguments. Another potential line of work is to generalize our results for arbitrary data models. However, since our arguments rely heavily on the concentration of measure, it is hard to extend the analysis to arbitrary distributions. Therefore, we require distribution-agnostic tools.

## 6 ACKNOWLEDGEMENTS

K.F. would like to acknowledge the support of the Natural Sciences and Engineering Research Council of Canada (NSERC). Cette recherche a été financée par le Conseil de recherches en sciences naturelles et en génie du Canada (CRSNG), [RGPIN-2019-04067, DGECR-2019-00147]. A.J. acknowledges the support of the Natural Sciences and Engineering Research Council of Canada (NSERC). Cette recherche a été financée par le Conseil de recherches en sciences naturelles et en génie du Canada (CRSNG), [RGPIN-2020-04597, DGECR-2020-00199].

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

## A   PROOFS

### A.1   ASSUMPTIONS AND NOTATION

**Assumption 1.** *For the XOR-GMM data model, the means of the Gaussian mixture are such that $\langle \boldsymbol{\mu}, \boldsymbol{\nu} \rangle = 0$ and $\|\boldsymbol{\mu}\|_2 = \|\boldsymbol{\nu}\|_2$.*

We denote $[x]_+ = \mathrm{ReLU}(x)$ and $\varphi(x) = \mathrm{sigmoid}(x) = 1/1+e^{-x}$, applied element-wise on the inputs. For any vector $\mathbf{v}$, $\hat{\mathbf{v}} = \frac{\mathbf{v}}{\|\mathbf{v}\|_2}$ denotes the normalized $\mathbf{v}$. We use $\gamma = \|\boldsymbol{\mu} - \boldsymbol{\nu}\|_2$ to denote the distance between the means of the inter-class components of the mixture model, and $\gamma'$ to denote the norm of the means, $\gamma' = \gamma/\sqrt{2} = \|\boldsymbol{\mu}\|_2 = \|\boldsymbol{\nu}\|_2$.

Given intra-class and inter-class edge probabilities $p$ and $q$, we define $\Gamma(p, q) = \frac{|p-q|}{p+q}$. We denote the probability density function of a standard Gaussian by $\phi(x)$, and the cumulative distribution function by $\Phi(x)$. The complementary distribution function is denoted by $\Phi_{\mathrm{c}}(x) = 1 - \Phi(x)$.

### A.2   ELEMENTARY RESULTS

In this section, we state preliminary results about the concentration of the degrees of all nodes and the number of common neighbours for all pairs of nodes, along with the effects of a graph convolution on the mean and the variance of some data. Our results regarding the merits of graph convolutions rely heavily on these arguments.

**Proposition A.1** (Concentration of degrees). *Assume that the graph density is $p, q = \Omega(\frac{\log^2 n}{n})$. Then for any constant $c > 0$, with probability at least $1 - 2n^{-c}$, we have for all $i \in [n]$ that*

$$\mathbf{deg}(i) = \frac{n}{2}(p + q)(1 \pm o_n(1)), \qquad \frac{1}{\mathbf{deg}(i)} = \frac{2}{n(p + q)}(1 \pm o_n(1)),$$

$$\frac{1}{\mathbf{deg}(i)}\left(\sum_{j \in C_1} a_{ij} - \sum_{j \in C_0} a_{ij}\right) = (2\varepsilon_i - 1)\frac{p - q}{p + q}(1 + o_n(1)),$$

*where the error term $o_n(1) = O\left(\sqrt{\frac{c}{\log n}}\right)$.*

*Proof.* Note that $\mathbf{deg}(i)$ is a sum of $n$ Bernoulli random variables, hence, we have by the Chernoff bound (Vershynin, 2018, Section 2) that

$$\mathbf{Pr}\left[\mathbf{deg}(i) \in \left[\frac{n}{2}(p+q)(1-\delta), \frac{n}{2}(p+q)(1+\delta)\right]^{\mathsf{c}}\right] \leq 2\exp(-Cn(p+q)\delta^2),$$

for some $C > 0$. We now choose $\delta = \sqrt{\frac{(c+1)\log n}{Cn(p+q)}}$ for a large constant $c > 0$. Note that since $p, q = \Omega(\log^2 n/n)$, we have that $\delta = O(\sqrt{\frac{c}{\log n}}) = o_n(1)$. Then following a union bound over $i \in [n]$, we obtain that with probability at least $1 - 2n^{-c}$,

$$\mathbf{deg}(i) = \frac{n}{2}(p+q)\left(1 \pm O\left(\sqrt{\frac{c}{\log n}}\right)\right) \quad \text{for all } i \in [n],$$

$$\frac{1}{\mathbf{deg}(i)} = \frac{2}{n(p+q)}\left(1 \pm O\left(\sqrt{\frac{c}{\log n}}\right)\right) \quad \text{for all } i \in [n].$$

Note that $\frac{1}{\mathbf{deg}(i)} \sum_{j \in C_b} a_{ij}$ for any $b \in \{0, 1\}$ is a sum of independent Bernoulli random variables. Hence, by a similar argument, we have that with probability at least $1 - 2n^{-c}$,

$$\frac{1}{\mathbf{deg}(i)}\left(\sum_{j \in C_1} a_{ij} - \sum_{j \in C_0} a_{ij}\right) = (2\varepsilon_i - 1)\frac{p-q}{p+q}(1 + o_n(1)) \text{ for all } i \in [n]. \qquad \square$$

**Proposition A.2** (Concentration of the number of common neighbours). *Assume that the graph density is $p, q = \Omega(\frac{\log n}{\sqrt{n}})$. Then for any constant $c > 0$, with probability at least $1 - 2n^{-c}$,*

$$|N_i \cap N_j| = \frac{n}{2}(p^2 + q^2)(1 \pm o_n(1)) \qquad \text{for all } i \sim j,$$

$$|N_i \cap N_j| = npq(1 \pm o_n(1)) \qquad \text{for all } i \nsim j,$$

*where the error term $o_n(1) = O\left(\sqrt{\frac{c}{\log n}}\right)$.*

*Proof.* For any two distinct nodes $i, j \in [n]$ we have that the number of common neighbours of $i$ and $j$ is $|N_i \cap N_j| = \sum_{k \in [n]} a_{ik} a_{jk}$. This is a sum of independent Bernoulli random variables, with mean $\mathbb{E}|N_i \cap N_j| = \frac{n}{2}(p^2 + q^2)$ for $i \sim j$ and $\mathbb{E}|N_i \cap N_j| = npq$ for $i \nsim j$. Denote $\mu_{ij} = \mathbb{E}|N_i \cap N_j|$. Therefore, by the Chernoff bound (Vershynin, 2018, Section 2), we have for a fixed pair of nodes $(i, j)$ that

$$\mathbf{Pr}\left[|N_i \cap N_j| \in [\mu_{ij}(1 - \delta_{ij}), \mu_{ij}(1 + \delta_{ij})]^{\mathsf{c}}\right] \leq 2\exp(-C\mu_{ij}\delta_{ij}^2)$$

for some constant $C > 0$. We now choose $\delta_{ij} = \sqrt{\frac{(c+2)\log n}{C\mu_{ij}}}$ for any large $c > 0$. Note that since $p, q = \Omega(\log n/\sqrt{n})$, we have that $\delta_{ij} = O(\sqrt{\frac{c}{\log n}}) = o_n(1)$. Then following a union bound over all pairs $(i, j) \in [n] \times [n]$, we obtain that with probability at least $1 - 2n^{-c}$, for all pairs of nodes $(i, j)$ we have

$$|N_i \cap N_j| = \frac{n}{2}(p^2 + q^2)(1 \pm o_n(1)) \qquad \text{for all } i \sim j,$$

$$|N_i \cap N_j| = npq(1 \pm o_n(1)) \qquad \text{for all } i \nsim j. \qquad \square$$

**Lemma A.3** (Variance reduction). *Denote the event from Proposition A.1 to be $B$. Let $\{\mathbf{X}_i\}_{i \in [n]} \in \mathbb{R}^{n \times d}$ be an iid sample of data. For a graph with adjacency matrix $\mathbf{A}$ (including self-loops) and a fixed integer $K > 0$, define a $K$-convolution to be $\tilde{\mathbf{X}} = (\mathbf{D}^{-1}\mathbf{A})^K \mathbf{X}$. Then we have*

$$\mathbf{Cov}(\tilde{\mathbf{X}}_i \mid B) = \rho(K)\mathbf{Cov}(\mathbf{X}_i), \text{ where } \rho(K) = \left(\frac{1 + o_n(1)}{\Delta}\right)^{2K} \sum_{j \in [n]} \mathbf{A}^K(i, j)^2.$$

*Here, $\mathbf{A}^K(i, j)$ is the entry in the $i$th row and $j$th column of the exponentiated matrix $\mathbf{A}^K$ and $\Delta = \mathbb{E}\,\mathbf{deg} = \frac{n}{2}(p + q)$.*

*Proof.* For a matrix $\mathbf{M}$, the $i$th convolved data point is $\tilde{\mathbf{X}}_i = \mathbf{M}_i^\top \mathbf{X}$, where $\mathbf{M}_i^\top$ denotes the $i$th row of $\mathbf{M}$. Since $\mathbf{X}_i$ are iid, we have

$$\mathbf{Cov}(\tilde{\mathbf{X}}_i) = \sum_{j \in [n]} (\mathbf{M}_{ij})^2 \mathbf{Cov}(\mathbf{X}_j).$$

It remains to compute the entries of the matrix $\mathbf{M} = (\mathbf{D}^{-1}\mathbf{A})^K$. Note that we have $\mathbf{D}^{-1}\mathbf{A}(i,j) = a_{ij}/\mathbf{deg}(i)$, so we obtain that

$$\mathbf{M}_{ij} = (\mathbf{D}^{-1}\mathbf{A})^K(i,j) = \sum_{j_1=1}^{n} \sum_{j_2=1}^{n} \cdots \sum_{j_{K-1}=1}^{n} \frac{a_{ij_1} a_{j_1 j_2} \cdots a_{j_{K-2}j_{K-1}} a_{j_{K-1}j}}{\mathbf{deg}(i)\mathbf{deg}(j_1)\cdots\mathbf{deg}(j_{K-1})}.$$

Recall that on the event $B$, the degrees of all nodes are $\Delta(1 \pm o_n(1))$, and hence, we have that

$$\mathbf{M}_{ij} = \frac{(1 \pm o_n(1))^K}{\Delta^K} \sum_{j_1=1}^{n} \sum_{j_2=1}^{n} \cdots \sum_{j_{K-1}=1}^{n} a_{ij_1} \cdots a_{j_{K-2}j_{K-1}} a_{j_{K-1}j},$$

where the error $o_n(1) = O(\frac{1}{\sqrt{\log n}})$. The sum of these products of the entries of $\mathbf{A}$ is simply the number of length-$K$ paths from node $i$ to $j$, i.e., $\mathbf{A}^K(i,j)$. Thus, we have

$$\mathbf{Cov}(\tilde{\mathbf{X}}_i \mid B) = \sum_{j \in [n]} (\mathbf{M}_{ij})^2 \mathbf{Cov}(\mathbf{X}_j) = \left( \frac{1 + o_n(1)}{\Delta} \right)^{2K} \sum_{j \in [n]} \mathbf{A}^K(i,j)^2 \mathbf{Cov}(\mathbf{X}_j).$$

Since $\mathbf{X}_j$ are iid, we obtain that $\rho(K) = \left( \frac{1 + o_n(1)}{\Delta} \right)^{2K} \sum_{j \in [n]} \mathbf{A}^K(i,j)^2$. $\square$

Let us briefly discuss the implications of Lemma A.3. Consider a sample $(\mathbf{A}, \mathbf{X})$ drawn from XOR-CSBM$(n, d, \boldsymbol{\mu}, \boldsymbol{\nu}, \sigma^2, p, q)$ for the symmetric case where exactly $n/2$ nodes are in each of the two classes. We have that

$$\mathbb{E}\mathbf{A} = \begin{pmatrix} p\mathbf{I}_{n/2} & q\mathbf{I}_{n/2} \\ q\mathbf{I}_{n/2} & p\mathbf{I}_{n/2} \end{pmatrix}.$$

This gives us $\mathbb{E}\rho(K) \approx \frac{1}{n}(1 + \Gamma(p,q)^{2K})$ for any $K \geq 2$. Recall that a single graph convolution reduces the distance between the means by a factor of $\Gamma(p,q)$. Hence, to comment on the performance of an arbitrary number of convolutions, $K$, we might hope to compare the reduction in this distance, $\Gamma(p,q)^K$ with the reduction in the variance $(\rho(K))$ to obtain a condition on $K$ in terms of $n$, $p$, and $q$. The challenge, however, lies in the fact that in deeper layers, computing $\rho(K)$ is non-trivial due to node features being highly correlated. Moreover, an argument to claim that $\rho(K) \approx \mathbb{E}\rho(K)$ is needed for this approach, which seems to require strong density assumptions on the graph.

We now state a result about the output of the (Bayes) optimal classifier for the XOR-GMM data model that is used in several of our proofs.

**Lemma A.4.** *Let* $h(\boldsymbol{x}) = |\langle \boldsymbol{x}, \hat{\boldsymbol{\nu}} \rangle| - |\langle \boldsymbol{x}, \hat{\boldsymbol{\mu}} \rangle|$ *for all* $\boldsymbol{x} \in \mathbb{R}^d$ *and define*

$$\zeta(t) = t \operatorname{erf}(t) - \frac{1}{\sqrt{\pi}}\left(1 - e^{-t^2}\right).$$

*Then we have*

1. *The expectation* $\mathbb{E}h(\mathbf{X}_i) = \begin{cases} -\sqrt{2}\sigma\zeta(\gamma/2\sigma) & i \in C_0 \\ \sqrt{2}\sigma\zeta(\gamma/2\sigma) & i \in C_1 \end{cases}$.

2. *For any* $\gamma, \sigma > 0$ *such that* $\gamma = \Omega_n(\sigma)$, *we have that* $\zeta(\frac{\gamma}{\sigma}) = \Omega(\frac{\gamma}{\sigma})$.

3. *For any* $\gamma, \sigma > 0$ *such that* $\gamma = o_n(\sigma)$, *we have that* $\zeta(\frac{\gamma}{\sigma}) = \Omega(\frac{\gamma^2}{\sigma^2})$.

*Proof.* For part one, observe that $\langle \mathbf{X}_i, \hat{\boldsymbol{\mu}} \rangle$ and $\langle \mathbf{X}_i, \hat{\boldsymbol{\nu}} \rangle$ are Gaussian random variables with variance $\sigma^2$ and means $\gamma/\sqrt{2}, 0$ if $\varepsilon_i = 0$ and $0, \gamma/\sqrt{2}$ if $\varepsilon_i = 1$, respectively. Thus, $|\langle \mathbf{X}_i, \hat{\boldsymbol{\mu}} \rangle|$ and $|\langle \mathbf{X}_i, \hat{\boldsymbol{\nu}} \rangle|$ are folded-Gaussian random variables and we have $\mathbb{E}h(\mathbf{X}_i) = -\sqrt{2}\zeta(\gamma/\sqrt{2}\sigma)$ if $i \in C_0$ and $\mathbb{E}h(\mathbf{X}_i) = \sqrt{2}\zeta(\gamma/\sqrt{2}\sigma)$ otherwise.

We now write

$$\zeta(t) = t\left(\text{erf}(t) - \frac{1}{t\sqrt{\pi}}(1 - e^{-t^2})\right) = tH(t),$$

where $H(t) = \text{erf}(t) - 1/t\sqrt{\pi}(1 - e^{-t^2})$.

For part two, note that $H(t)$ is an increasing function in the range $[-1, 1]$ and $H(t) > 0$ for $t > 0$. Hence, for $t \geq C$ for some positive constant $C$, $H(t) \geq H(C) = C'$, therefore, $\zeta(t) = tH(t) \geq C't$.

For part three when $t = o_n(1)$, we use the series expansion of $h(t)$ about $t = 0$ to obtain that

$$h(t) = \frac{t}{\sqrt{\pi}} - \frac{t^3}{6\sqrt{\pi}} + O(t^5) \geq \frac{t}{\sqrt{\pi}} - \frac{t^3}{6\sqrt{\pi}} = \Omega(t).$$

Hence, $\zeta(t) = th(t) = \Omega(t^2)$. Putting $t = \gamma/\sigma$ completes the proof. $\square$

**Fact A.5.** *For any $x \in [0, 1]$, $\frac{x}{2} \leq \log(1 + x) \leq x$.*

### A.3 PROOF OF THEOREM 1 PART ONE

In this section we prove our first result about the fraction of misclassified points in the absence of graphical information. We begin by computing the Bayes optimal classifier for the data model XOR-GMM (see Section 2.1). A Bayes classifier, denoted by $h^*(\boldsymbol{x})$, maximizes the posterior probability of observing a label given the input data $\boldsymbol{x}$. More precisely, $h^*(\boldsymbol{x}) = \text{argmax}_{b \in \{0,1\}} \mathbf{Pr}[y = b \mid \mathbf{x} = \boldsymbol{x}]$, where $\boldsymbol{x} \in \mathbb{R}^d$ represents a single data point.

**Lemma A.6.** *For some fixed $\boldsymbol{\mu}, \boldsymbol{\nu} \in \mathbb{R}^d$ and $\sigma^2 > 0$, the Bayes optimal classifier, $h^*(\boldsymbol{x}) : \mathbb{R}^d \to \{0, 1\}$ for the data model XOR-GMM$(n, d, \boldsymbol{\mu}, \boldsymbol{\nu}, \sigma^2)$ is given by*

$$h^*(\boldsymbol{x}) = \mathbb{1}(|\langle \boldsymbol{x}, \boldsymbol{\mu}\rangle| < |\langle \boldsymbol{x}, \boldsymbol{\nu}\rangle|) = \begin{cases} 0 & |\langle \boldsymbol{x}, \boldsymbol{\mu}\rangle| \geq |\langle \boldsymbol{x}, \boldsymbol{\nu}\rangle| \\ 1 & |\langle \boldsymbol{x}, \boldsymbol{\mu}\rangle| < |\langle \boldsymbol{x}, \boldsymbol{\nu}\rangle| \end{cases},$$

*where $\mathbb{1}$ is the indicator function.*

*Proof.* Note that $\mathbf{Pr}[y = 0] = \mathbf{Pr}[y = 1] = \frac{1}{2}$. Let $f_{\mathbf{x}}(\boldsymbol{x})$ denote the density function of a continuous random vector $\mathbf{x}$. Therefore, for any $b \in \{0, 1\}$,

$$\mathbf{Pr}[y = b \mid \mathbf{x} = \boldsymbol{x}] = \frac{\mathbf{Pr}[y = b]\, f_{\mathbf{x}|y}(\boldsymbol{x} \mid y = b)}{\sum_{c \in \{0,1\}} \mathbf{Pr}[y = c]\, f_{\mathbf{x}|y}(\boldsymbol{x} \mid y = c)} = \frac{1}{1 + \frac{f_{\mathbf{x}|y}(\boldsymbol{x}|y=1-b)}{f_{\mathbf{x}|y}(\boldsymbol{x}|y=b)}}.$$

Let's compute this for $b = 0$. We have

$$\frac{f_{\mathbf{x}|y}(\boldsymbol{x} \mid y = 1)}{f_{\mathbf{x}|y}(\boldsymbol{x} \mid y = 0)} = \frac{\cosh(\langle \boldsymbol{x}, \boldsymbol{\nu}\rangle/\sigma^2)}{\cosh(\langle \boldsymbol{x}, \boldsymbol{\mu}\rangle/\sigma^2)} \exp\left(\frac{\|\boldsymbol{\mu}\|^2 - \|\boldsymbol{\nu}^2\|}{2\sigma^2}\right) = \frac{\cosh(\langle \boldsymbol{x}, \boldsymbol{\nu}\rangle/\sigma^2)}{\cosh(\langle \boldsymbol{x}, \boldsymbol{\mu}\rangle/\sigma^2)},$$

where in the last equation we used the assumption that $\|\boldsymbol{\mu}\| = \|\boldsymbol{\nu}\|$. The decision regions are then identified by: $\mathbf{Pr}[y = 0 \mid \mathbf{x}] \geq 1/2$ for label 0 and $\mathbf{Pr}[y = 0 \mid \mathbf{x}] < 1/2$ for label 1.

Thus, for label 0, we need $\frac{f_{\mathbf{x}|y}(\boldsymbol{x}|y=1)}{f_{\mathbf{x}|y}(\boldsymbol{x}|y=0)} < 1$, which implies that $\frac{\cosh(\langle \boldsymbol{x}, \boldsymbol{\nu}\rangle/\sigma^2)}{\cosh(\langle \boldsymbol{x}, \boldsymbol{\mu}\rangle/\sigma^2)} \leq 1$. Now we note that $\cosh(x) \leq \cosh(y) \implies |x| \leq |y|$ for all $x, y \in \mathbb{R}$, hence, we have $|\langle \boldsymbol{x}, \boldsymbol{\mu}\rangle| \geq |\langle \boldsymbol{x}, \boldsymbol{\nu}\rangle|$. Similarly, we have the complementary condition for label 1. $\square$

Next, we design a two-layer and a three-layer network and show that for a particular choice of parameters $\theta = (\mathbf{W}^{(l)}, \mathbf{b}^{(l)})$ for $l \in \{1, 2\}$ for the two-layer case and $l \in \{1, 2, 3\}$ for the three-layer case, the networks realize the optimal classifier described in Lemma A.6.

**Proposition A.7.** *Consider two-layer and three-layer networks of the form described in Section 2.2, without biases (i.e., $\mathbf{b}^{(l)} = \mathbf{0}$ for all layers $l$), for parameters $\mathbf{W}^{(l)}$ and some $R \in \mathbb{R}^+$ as follows.*

*1. For the two-layer network,*

$$\mathbf{W}^{(1)} = R(\hat{\boldsymbol{\mu}} \quad -\hat{\boldsymbol{\mu}} \quad \hat{\boldsymbol{\nu}} \quad -\hat{\boldsymbol{\nu}}), \qquad \mathbf{W}^{(2)} = (-1 \quad -1 \quad 1 \quad 1)^\top.$$

2. *For the three-layer network,*

$$\mathbf{W}^{(1)} = R\left(\hat{\boldsymbol{\mu}} \quad -\hat{\boldsymbol{\mu}} \quad \hat{\boldsymbol{\nu}} \quad -\hat{\boldsymbol{\nu}}\right), \quad \mathbf{W}^{(2)} = \begin{pmatrix} -1 & 1 \\ -1 & 1 \\ 1 & -1 \\ 1 & -1 \end{pmatrix}, \quad \mathbf{W}^{(3)} = \begin{pmatrix} 1 \\ -1 \end{pmatrix}.$$

*Then for any $\sigma > 0$, the defined networks realize the Bayes optimal classifier for the data model XOR-GMM$(n, d, \boldsymbol{\mu}, \boldsymbol{\nu}, \sigma^2)$.*

*Proof.* Note that the output of the two-layer network is $\varphi([\mathbf{X}\mathbf{W}^{(1)}]_+\mathbf{W}^{(2)})$, which is interpreted as the probability with which the network believes that the input is in the class with label 1. The final prediction for the class label is thus assigned to be 1 if the output is $\geq 0.5$, and 0 otherwise. For each $i \in [n]$, we have that the output of the network on data point $i$ is

$$\hat{y}_i = \varphi(R(-[\langle \mathbf{X}_i, \hat{\boldsymbol{\mu}}\rangle]_+ - [-\langle \mathbf{X}_i, \hat{\boldsymbol{\mu}}\rangle]_+ + [\langle \mathbf{X}_i, \hat{\boldsymbol{\nu}}\rangle]_+ + [-\langle \mathbf{X}_i, \hat{\boldsymbol{\nu}}\rangle]_+))$$
$$= \varphi((R(|\langle \mathbf{X}_i, \hat{\boldsymbol{\nu}}\rangle| - |\langle \mathbf{X}_i, \hat{\boldsymbol{\mu}}\rangle|))),$$

where we used the fact that $[t]_+ + [-t]_+ = |t|$ for all $t \in \mathbb{R}$. Similarly, for the three-layer network, the output is $\varphi([[\mathbf{X}\mathbf{W}^{(1)}]_+\mathbf{W}^{(2)}]_+\mathbf{W}^{(3)})$. So we have for each $i \in [n]$ that

$$\hat{y}_i = \varphi\Bigg(R\Big([-[\langle \mathbf{X}_i, \hat{\boldsymbol{\mu}}\rangle]_+ - [-\langle \mathbf{X}_i, \hat{\boldsymbol{\mu}}\rangle]_+ + [\langle \mathbf{X}_i, \hat{\boldsymbol{\nu}}\rangle]_+ + [-\langle \mathbf{X}_i, \hat{\boldsymbol{\nu}}\rangle]_+]_+$$

$$- [[\langle \mathbf{X}_i, \hat{\boldsymbol{\mu}}\rangle]_+ + [-\langle \mathbf{X}_i, \hat{\boldsymbol{\mu}}\rangle]_+ - [\langle \mathbf{X}_i, \hat{\boldsymbol{\nu}}\rangle]_+ - [-\langle \mathbf{X}_i, \hat{\boldsymbol{\nu}}\rangle]_+]_+\Big)\Bigg)$$

$$= \varphi\left(R([|\langle \mathbf{X}_i, \hat{\boldsymbol{\nu}}\rangle| - |\langle \mathbf{X}_i, \hat{\boldsymbol{\mu}}\rangle|]_+ - [|\langle \mathbf{X}_i, \hat{\boldsymbol{\mu}}\rangle| - |\langle \mathbf{X}_i, \hat{\boldsymbol{\nu}}\rangle|]_+)\right)$$
$$= \varphi\left(R(|\langle \mathbf{X}_i, \hat{\boldsymbol{\nu}}\rangle| - |\langle \mathbf{X}_i, \hat{\boldsymbol{\mu}}\rangle|)\right),$$

where in the last equation we used the fact that $[t]_+ - [-t]_+ = t$ for all $t \in \mathbb{R}$.

The final prediction is then obtained by considering the maximum posterior probability among the class labels 0 and 1, and thus,

$$\text{pred}(\mathbf{X}_i) = \mathbb{1}(R\,|\langle \mathbf{X}_i, \hat{\boldsymbol{\mu}}\rangle| < R\,|\langle \mathbf{X}_i, \hat{\boldsymbol{\nu}}\rangle|) = \mathbb{1}(|\langle \mathbf{X}_i, \boldsymbol{\mu}\rangle| < |\langle \mathbf{X}_i, \boldsymbol{\nu}\rangle|),$$

which matches the Bayes classifier in Lemma A.6. $\qquad\square$

We now restate the relevant theorem below for convenience.

**Theorem** (Restatement of part one of Theorem 1)**.** *Let $\mathbf{X} \in \mathbb{R}^{n \times d} \sim$ XOR-GMM$(n, d, \boldsymbol{\mu}, \boldsymbol{\nu}, \sigma^2)$. Assume that $\|\boldsymbol{\mu} - \boldsymbol{\nu}\|_2 \leq K\sigma$ and let $h(\mathbf{x}) : \mathbb{R}^d \to \{0, 1\}$ be any binary classifier. Then for any $K > 0$ and any $\epsilon \in (0, 1)$, at least a fraction $2\Phi_c\left(\frac{K}{2}\right)^2 - O(n^{-\epsilon/2})$ of all data points are misclassified by $h$ with probability at least $1 - \exp(-2n^{1-\epsilon})$.*

*Proof.* Recall from Lemma A.6 that for successful classification, we require for every $i \in [n]$,

$$|\langle \mathbf{X}_i, \boldsymbol{\mu}\rangle| \geq |\langle \mathbf{X}_i, \boldsymbol{\nu}\rangle| \qquad i \in C_0,$$
$$|\langle \mathbf{X}_i, \boldsymbol{\mu}\rangle| < |\langle \mathbf{X}_i, \boldsymbol{\nu}\rangle| \qquad i \in C_1.$$

Let's try to upper bound the probability of the above event, i.e., the probability that the data is classifiable. We consider only class $C_0$, since the analysis for $C_1$ is symmetric and similar. For $i \in C_0$, we can write $\mathbf{X}_i = \boldsymbol{\mu} + \sigma\mathbf{g}_i$, where $\mathbf{g}_i \sim \mathcal{N}(\mathbf{0}, I)$. Recall that $\gamma = \|\boldsymbol{\mu} - \boldsymbol{\nu}\|_2$ and $\gamma' = \gamma/\sqrt{2} = \|\boldsymbol{\mu}\|_2 = \|\boldsymbol{\nu}\|_2$. Then we have for any fixed $i \in C_0$ that

$$\mathbf{Pr}\left[|\langle \mathbf{X}_i, \boldsymbol{\mu}\rangle| \geq |\langle \mathbf{X}_i, \boldsymbol{\nu}\rangle|\right] = \mathbf{Pr}\left[|\gamma' + \sigma\,\langle \mathbf{g}_i, \hat{\boldsymbol{\mu}}\rangle| \geq |\sigma\,\langle \mathbf{g}_i, \hat{\boldsymbol{\nu}}\rangle|\right]$$
$$\leq \mathbf{Pr}\left[\gamma' + \sigma\,|\langle \mathbf{g}_i, \hat{\boldsymbol{\mu}}\rangle| \geq \sigma\,|\langle \mathbf{g}_i, \hat{\boldsymbol{\nu}}\rangle|\right] \quad \text{(by triangle inequality)}$$
$$\leq \mathbf{Pr}\left[|\langle \mathbf{g}_i, \hat{\boldsymbol{\nu}}\rangle| - |\langle \mathbf{g}_i, \hat{\boldsymbol{\mu}}\rangle| \leq {}^{K}\!/\!\sqrt{2}\right] \quad \text{(using } \gamma \leq K\sigma\text{)}.$$

We now define random variables $Z_1 = \langle \mathbf{g}_i, \hat{\boldsymbol{\nu}} \rangle$ and $Z_2 = \langle \mathbf{g}_i, \hat{\boldsymbol{\mu}} \rangle$ and note that $Z_1, Z_2 \sim \mathcal{N}(0,1)$ and $\mathbb{E}[Z_1 Z_2] = 0$. Let $K' = K/\sqrt{2}$. We now have

$$
\begin{aligned}
\mathbf{Pr}\left[|Z_1| - |Z_2| \le K'\right] &= 4\mathbf{Pr}\left[Z_1 - Z_2 \le K', \, Z_1, Z_2 \ge 0\right] \\
&= 4\int_0^\infty \mathbf{Pr}\left[0 \le Z_1 \le z + K'\right]\phi(z)dz \\
&= 4\int_0^\infty \left(\Phi(z + K') - \frac{1}{2}\right)\phi(z)dz = 4\int_0^\infty \Phi(z + K')\,\phi(z)dz - 1 \\
&= 2\Phi(K/2) + 2\Phi(K/2)\Phi_{\rm c}(K/2) - 1 = 1 - 2\Phi_{\rm c}(K/2)^2.
\end{aligned}
$$

To evaluate the integral above, we used (Owen, 1980, Table 1:10,010.6 and Table 2:2.3). Thus, the probability that a point $i \in C_0$ is misclassified is lower bounded as follows

$$
\mathbf{Pr}\left[\mathbf{X}_i \text{ is misclassified}\right] \ge 2\Phi_{\rm c}\left(K/2\right)^2 = \tau_K.
$$

Note that this is a decreasing function of $K$, implying that the probability of misclassification decreases as we increase the distance between the means, and is maximum for $K = 0$.

Define $M(n)$ for a fixed $K$ to be the fraction of misclassified nodes in $C_0$. Define $x_i$ to be the indicator random variable $\mathbb{1}(\mathbf{X}_i \text{ is misclassified})$. Then $x_i$ are Bernoulli random variables with mean at least $\tau_K$, and $\mathbb{E}M(n) = \frac{2}{n}\sum_{i \in C_0}\mathbb{E}x_i \ge \tau_K$. Using Hoeffding's inequality (Vershynin, 2018, Theorem 2.2.6), we have that for any $t > 0$,

$$
\mathbf{Pr}\left[M(n) \ge \tau_K - t\right] \ge \mathbf{Pr}\left[M(n) \ge \mathbb{E}M(n) - t\right] \ge 1 - \exp(-nt^2).
$$

Choosing $t = n^{-\epsilon/2}$ for any $\epsilon \in (0,1)$ yields

$$
\mathbf{Pr}\left[M(n) \ge \tau_K - n^{-\epsilon/2}\right] \ge 1 - \exp(-n^{1-\epsilon}). \qquad \square
$$

### A.4 Proof of Theorem 1 part two

In this section, we show that in the positive regime (sufficiently large distance between the means), there exists a two-layer MLP that obtains an arbitrarily small loss, and hence, successfully classifies a sample drawn from the XOR-GMM model with overwhelming probability.

**Theorem** (Restatement of part two of Theorem 1). *Let $\mathbf{X} \in \mathbb{R}^{n \times d} \sim$ XOR-GMM$(n, d, \boldsymbol{\mu}, \boldsymbol{\nu}, \sigma^2)$. For any $\epsilon > 0$, if the distance between the means is $\|\boldsymbol{\mu} - \boldsymbol{\nu}\|_2 = \Omega(\sigma(\log n)^{\frac{1}{2}+\epsilon})$, then for any $c > 0$, with probability at least $1 - O(n^{-c})$, the two-layer and three-layer networks described in Proposition A.7 classify all data points, and obtain a cross-entropy loss given by*

$$
\ell_\theta(\mathbf{X}) = C\exp\left(-\frac{R}{\sqrt{2}}\|\boldsymbol{\mu} - \boldsymbol{\nu}\|_2\left(1 \pm \sqrt{c}/(\log n)^\epsilon\right)\right),
$$

*where $C \in [1/2, 1]$ is an absolute constant.*

*Proof.* Consider the two-layer and three-layer MLPs described in Proposition A.7, for which we have $\hat{y}_i = \varphi\left(R(|\langle \mathbf{X}_i, \hat{\boldsymbol{\nu}} \rangle| - |\langle \mathbf{X}_i, \hat{\boldsymbol{\mu}} \rangle|)\right)$. We now look at the loss for a single data point $\mathbf{X}_i$,

$$
\begin{aligned}
\ell_i(\mathbf{X}, \theta) &= -y_i \log(\hat{y}_i) - (1 - y_i)\log(1 - \hat{y}_i) \\
&= \log\left(1 + \exp\left((1 - 2y_i)R(|\langle \mathbf{X}_i, \hat{\boldsymbol{\nu}} \rangle| - |\langle \mathbf{X}_i, \hat{\boldsymbol{\mu}} \rangle|)\right)\right).
\end{aligned}
$$

Note that $\langle \mathbf{X}_i - \mathbb{E}\mathbf{X}_i, \hat{\boldsymbol{\mu}} \rangle$ and $\langle \mathbf{X}_i - \mathbb{E}\mathbf{X}_i, \hat{\boldsymbol{\nu}} \rangle$ are mean $0$ Gaussian random variables with variance $\sigma^2$. So for any fixed $i \in [n]$ and $\mathbf{m}_c \in \{\boldsymbol{\mu}, \boldsymbol{\nu}\}$, we use (Vershynin, 2018, Proposition 2.1.2) to obtain

$$
\mathbf{Pr}\left[|\langle \mathbf{X}_i - \mathbb{E}\mathbf{X}_i, \hat{\mathbf{m}}_c \rangle| > t\right] \le \frac{\sigma}{t\sqrt{2\pi}}\exp\left(-\frac{t^2}{2\sigma^2}\right).
$$

Then by a union bound over all $i \in [n]$ and $\mathbf{m}_c \in \{\boldsymbol{\mu}, \boldsymbol{\nu}\}$, we have that

$$
\mathbf{Pr}\left[|\langle \mathbf{X}_i - \mathbb{E}\mathbf{X}_i, \hat{\mathbf{m}}_c \rangle| \le t \; \forall i \in [n], \; \mathbf{m}_c \in \{\boldsymbol{\mu}, \boldsymbol{\nu}\}\right] \ge 1 - \frac{n\sigma}{t}\sqrt{\frac{2}{\pi}}\exp\left(-\frac{t^2}{2\sigma^2}\right).
$$

We now set $t = \sigma\sqrt{2(c+1)\log n}$ for any large constant $c > 0$. We now have with probability at least $1 - \frac{n^{-c}}{\sqrt{\pi(c+1)\log n}}$ that

$$\langle \mathbf{X}_i, \hat{\boldsymbol{\mu}} \rangle = \langle \mathbb{E}\mathbf{X}_i, \hat{\boldsymbol{\mu}} \rangle \pm O(\sigma\sqrt{c\log n}), \quad \langle \mathbf{X}_i, \hat{\boldsymbol{\nu}} \rangle = \langle \mathbb{E}\mathbf{X}_i, \hat{\boldsymbol{\nu}} \rangle \pm O(\sigma\sqrt{c\log n}) \; \forall i \in [n].$$

Thus, we can write

$$\langle \mathbf{X}_i, \hat{\boldsymbol{\mu}} \rangle = \gamma'\left(1 \pm O\left(\sqrt{\frac{c}{\log n}}\right)\right), \qquad \langle \mathbf{X}_i, \hat{\boldsymbol{\nu}} \rangle = \gamma' \cdot O\left(\sqrt{\frac{c}{\log n}}\right) \qquad \forall i \in C_0, \quad (2)$$

$$\langle \mathbf{X}_i, \hat{\boldsymbol{\mu}} \rangle = \gamma' \cdot O\left(\sqrt{\frac{c}{\log n}}\right), \qquad \langle \mathbf{X}_i, \hat{\boldsymbol{\nu}} \rangle = \gamma'\left(1 \pm O\left(\sqrt{\frac{c}{\log n}}\right)\right) \quad \forall i \in C_1. \quad (3)$$

Using Eqs. (2) and (3) in the expression for the loss, we obtain for all $i \in [n]$,

$$\ell_i(\mathbf{X}, \theta) = \log(1 + \exp(-R\gamma'(1 \pm o_n(1)))),$$

where the error term $o_n(1) = \sqrt{c/\log n}$. The total loss is then given by

$$\ell_\theta(\mathbf{X}) = \frac{1}{n}\sum \ell_i(\mathbf{X}, \theta) = \log(1 + \exp(-R\gamma'(1 + o_n(1)))).$$

Next, Fact A.5 implies that for $t < 0$, $e^t/2 \le \log(1 + e^t) \le e^t$, hence, we have that there exists a constant $C \in [1/2, 1]$ such that

$$\ell_\theta(\mathbf{X}) = C \exp\left(-R\gamma'(1 + o_n(1))\right).$$

Note that by scaling the optimality constraint $R$, the loss can go arbitrarily close to 0. $\qquad \square$

## A.5 Graph convolution in the first layer

In this section, we show precisely why a graph convolution operation in the first layer is detrimental to the classification task.

**Proposition A.8.** *Fix a positive integer $d > 0$, $\sigma \in \mathbb{R}^+$ and $\boldsymbol{\mu}, \boldsymbol{\nu} \in \mathbb{R}^d$. Let $(\mathbf{A}, \mathbf{X}) \sim$ XOR-CSBM$(n, d, \boldsymbol{\mu}, \boldsymbol{\nu}, \sigma^2, p, q)$. Define $\tilde{\mathbf{X}}$ to be the transformed data after applying a graph convolution on $\mathbf{X}$, i.e., $\tilde{\mathbf{X}} = \mathbf{D}^{-1}\mathbf{A}\mathbf{X}$. Then in the regime where $p, q = \Omega(\frac{\log^2 n}{n})$, with probability at least $1 - 1/\text{poly}(n)$ we have that*

$$\mathbb{E}\tilde{\mathbf{X}}_i = \begin{cases} \dfrac{p\boldsymbol{\mu} + q\boldsymbol{\nu}}{2(p+q)} \cdot o_n(1) & i \in C_0 \\[2mm] \dfrac{p\boldsymbol{\nu} + q\boldsymbol{\mu}}{2(p+q)} \cdot o_n(1) & i \in C_1 \end{cases}.$$

*Hence, the distance between the means of the convolved data, given by $\frac{p-q}{2(p+q)}\|\boldsymbol{\mu} - \boldsymbol{\nu}\|_2 \cdot o_n(1)$ diminishes to 0 for $n \to \infty$.*

*Proof.* Fix $\boldsymbol{\mu}, \boldsymbol{\nu} \in \mathbb{R}^d$ and define the following sets:

$$C_{-\boldsymbol{\mu}} = \{i \mid \varepsilon_i = 0, \eta_i = 0\}, \qquad C_{\boldsymbol{\mu}} = \{i \mid \varepsilon_i = 0, \eta_i = 1\},$$
$$C_{-\boldsymbol{\nu}} = \{i \mid \varepsilon_i = 1, \eta_i = 0\}, \qquad C_{\boldsymbol{\nu}} = \{i \mid \varepsilon_i = 1, \eta_i = 1\}.$$

Denote $\tilde{\mathbf{X}} = \mathbf{D}^{-1}\mathbf{A}\mathbf{X}$ and note that for any $i \in [n]$, the row vector

$$\tilde{\mathbf{X}}_i = \frac{1}{\mathbf{deg}(i)}\sum_{j \in [n]} a_{ij}\mathbf{X}_j = \frac{1}{\mathbf{deg}(i)}\sum_{j \in [n]} a_{ij}(\mathbb{E}\mathbf{X}_j + \sigma\mathbf{g}_j)$$

$$= \frac{1}{\mathbf{deg}(i)}\left[\boldsymbol{\mu}\left(\sum_{j \in C_{\boldsymbol{\mu}}} a_{ij} - \sum_{j \in C_{-\boldsymbol{\mu}}} a_{ij}\right) + \boldsymbol{\nu}\left(\sum_{j \in C_{\boldsymbol{\nu}}} a_{ij} - \sum_{j \in C_{-\boldsymbol{\nu}}} a_{ij}\right) + \sigma\sum_{j \in [n]} a_{ij}\mathbf{g}_j\right],$$

where we used the fact that $\mathbf{X}_j = (2\eta_j - 1)((1 - \varepsilon_j)\boldsymbol{\mu} + \varepsilon_j\boldsymbol{\nu} + \sigma\mathbf{g}_j)$ for a set of iid Gaussian random vectors $\mathbf{g}_j \sim \mathcal{N}(\mathbf{0}, \mathbf{I}_d)$.

Note that since $\epsilon_i, \eta_i$ are Bernoulli random variables, using the Chernoff bound (Vershynin, 2018, Section 2), we have that with probability at least $1 - 1/\text{poly}(n)$,

$$|C_{-\boldsymbol{\mu}}| = |C_{\boldsymbol{\mu}}| = |C_{-\boldsymbol{\nu}}| = |C_{\boldsymbol{\nu}}| = \frac{n}{4}(1 \pm o_n(1)).$$

We now use an argument similar to Proposition A.1 to obtain that for any $c > 0$, with probability at least $1 - O(n^{-c})$, the following holds for all $i \in [n]$:

$$\frac{1}{\mathbf{deg}(i)}\left(\sum_{j \in C_{\boldsymbol{\mu}}} a_{ij} - \sum_{j \in C_{-\boldsymbol{\mu}}} a_{ij}\right) = O\left(\frac{(1-\varepsilon_i)p + \varepsilon_i q}{2(p+q)}\sqrt{\frac{c}{\log n}}\right),$$

$$\frac{1}{\mathbf{deg}(i)}\left(\sum_{j \in C_{\boldsymbol{\nu}}} a_{ij} - \sum_{j \in C_{-\boldsymbol{\nu}}} a_{ij}\right) = O\left(\frac{\varepsilon_i p + (1-\varepsilon_i)q}{2(p+q)}\sqrt{\frac{c}{\log n}}\right).$$

Hence, we have that for all $i \in [n]$,

$$\mathbb{E}\tilde{\mathbf{X}}_i = \left[\left(\frac{(1-\varepsilon_i)p + \varepsilon_i q}{2(p+q)}\right)\boldsymbol{\mu} + \left(\frac{\varepsilon_i p + (1-\varepsilon_i)q}{2(p+q)}\right)\boldsymbol{\nu}\right] \cdot O\left(\sqrt{\frac{c}{\log n}}\right)$$

$$= \begin{cases} \dfrac{p\boldsymbol{\mu} + q\boldsymbol{\nu}}{2(p+q)} \cdot o_n(1) & i \in C_0 \\ \dfrac{p\boldsymbol{\nu} + q\boldsymbol{\mu}}{2(p+q)} \cdot o_n(1) & i \in C_1 \end{cases}$$

Using the above result, we obtain the distance between the means, which is of the order $o_n(\gamma)$ and thus, diminishes to 0 as $n \to \infty$. $\qquad\square$

### A.6 PROOF OF THEOREM 2 PART ONE

We begin by computing the output of the network when one graph convolution is applied at any layer other than the first.

**Lemma A.9.** *Let $h(\boldsymbol{x}) = |\langle \boldsymbol{x}, \hat{\boldsymbol{\nu}}\rangle| - |\langle \boldsymbol{x}, \hat{\boldsymbol{\mu}}\rangle|$ for any $\boldsymbol{x} \in \mathbb{R}^d$. Consider the two-layer and three-layer networks in Proposition A.7 where the weight parameter of the last layer, $W^{(L)}$, is scaled by a factor of $\xi = \text{sgn}(p-q)$. If a graph convolution is added to these networks in either the second or the third layer then for a sample $(\mathbf{A}, \mathbf{X}) \sim XOR\text{-}CSBM(n, d, \boldsymbol{\mu}, \boldsymbol{\nu}, \sigma^2, p, q)$, the output of the networks for a point $i \in [n]$ is*

$$\hat{y}_i = \varphi(f_i^{(L)}(\mathbf{X})) = \varphi\left(\frac{R\,\text{sgn}(p-q)}{\mathbf{deg}(i)}\sum_{j \in [n]} a_{ij}h(\mathbf{X}_j)\right).$$

*Proof.* The networks with scaled parameters are given as follows.

1. For the two-layer network,

$$\mathbf{W}^{(1)} = R\left(\hat{\boldsymbol{\mu}} \quad -\hat{\boldsymbol{\mu}} \quad \hat{\boldsymbol{\nu}} \quad -\hat{\boldsymbol{\nu}}\right), \qquad \mathbf{W}^{(2)} = \xi\left(-1 \quad -1 \quad 1 \quad 1\right)^{\top}.$$

2. For the three-layer network,

$$\mathbf{W}^{(1)} = R\xi\left(\hat{\boldsymbol{\mu}} \quad -\hat{\boldsymbol{\mu}} \quad \hat{\boldsymbol{\nu}} \quad -\hat{\boldsymbol{\nu}}\right), \quad \mathbf{W}^{(2)} = \begin{pmatrix} -1 & 1 \\ -1 & 1 \\ 1 & -1 \\ 1 & -1 \end{pmatrix}, \quad \mathbf{W}^{(3)} = \xi\begin{pmatrix} 1 \\ -1 \end{pmatrix}.$$

When a graph convolution is applied at the second layer of this two-layer MLP, the output of the last layer for data $(\mathbf{A}, \mathbf{X})$ is $f_i^{(2)}(\mathbf{X}) = \mathbf{D}^{-1}\mathbf{A}[\mathbf{X}\mathbf{W}^{(1)}]_+\mathbf{W}^{(2)}$. Then we have

$$f_i^{(2)}(\mathbf{X}) = \frac{R\xi}{\mathbf{deg}(i)}\sum_{j \in [n]} a_{ij}(|\langle \mathbf{X}_j, \hat{\boldsymbol{\nu}}\rangle| - |\langle \mathbf{X}_j, \hat{\boldsymbol{\mu}}\rangle|) = \frac{R\xi}{\mathbf{deg}(i)}\sum_{j \in [n]} a_{ij}h(\mathbf{X}_j).$$

Similarly, when the graph convolution is applied at the second layer of the three-layer MLP, the output is $f_i^{(3)}(\mathbf{X}) = [\mathbf{D}^{-1}\mathbf{A}[\mathbf{X}\mathbf{W}^{(1)}]_+\mathbf{W}^{(2)}]_+\mathbf{W}^{(3)}$, and we have

$$f_i^{(3)}(\mathbf{X}) = \frac{R\xi}{\mathbf{deg}(i)}\left(\left[\sum_{j\in[n]} a_{ij}h(\mathbf{X}_j)\right]_+ - \left[-\sum_{j\in[n]} a_{ij}h(\mathbf{X}_j)\right]_+\right) = \frac{R\xi}{\mathbf{deg}(i)}\sum_{j\in[n]} a_{ij}h(\mathbf{X}_j).$$

Finally, when the graph convolution is applied at the third layer of the three-layer MLP, the output is $f_i^{(3)}(\mathbf{X}) = \mathbf{D}^{-1}\mathbf{A}[[\mathbf{X}\mathbf{W}^{(1)}]_+\mathbf{W}^{(2)}]_+\mathbf{W}^{(3)}$, and we have

$$f_i^{(3)}(\mathbf{X}) = \frac{R\xi}{\mathbf{deg}(i)}\sum_{j\in[n]} a_{ij}\left([|\langle\mathbf{X}_j,\hat{\boldsymbol{\nu}}\rangle| - |\langle\mathbf{X}_j,\hat{\boldsymbol{\mu}}\rangle|]_+ - [|\langle\mathbf{X}_j,\hat{\boldsymbol{\mu}}\rangle| - |\langle\mathbf{X}_j,\hat{\boldsymbol{\nu}}\rangle|]_+\right)$$

$$= \frac{R}{\mathbf{deg}(i)}\sum_{j\in[n]} a_{ij}(|\langle\mathbf{X}_j,\hat{\boldsymbol{\nu}}\rangle| - |\langle\mathbf{X}_j,\hat{\boldsymbol{\mu}}\rangle|) = \frac{R\xi}{\mathbf{deg}(i)}\sum_{j\in[n]} a_{ij}h(\mathbf{X}_j).$$

Therefore, in all cases where we have a single graph convolution, the output of the last layer is

$$f_i^{(L)}(\mathbf{X}) = \frac{R\operatorname{sgn}(p-q)}{\mathbf{deg}(i)}\sum_{j\in[n]} a_{ij}h(\mathbf{X}_j),$$

where $L\in\{2,3\}$ is the number of layers. $\qquad\square$

**Theorem** (Restatement of part one of Theorem 2). *Let $(\mathbf{A},\mathbf{X})\sim$ XOR-CSBM$(n,d,\boldsymbol{\mu},\boldsymbol{\nu},\sigma^2,p,q)$. Assume that $p,q = \Omega(\frac{\log^2 n}{n})$, and it holds that $\Gamma(p,q)\zeta(\gamma/2\sigma) = \omega\left(\sqrt{\frac{\log n}{n(p+q)}}\right)$, then for any $c > 0$, with probability at least $1 - O(n^{-c})$, the networks equipped with a graph convolution in the second or the third layer perfectly classify the data, and obtain the following loss:*

$$\ell_\theta(\mathbf{A},\mathbf{X}) = C'\exp\left(-C\sigma R\Gamma(p,q)\zeta(\gamma/2\sigma)(1\pm\sqrt{c/\log n})\right),$$

*where $C > 0$ and $C'\in[1/2,1]$ are constants and $R$ is the constraint from Eq. (1).*

*Proof.* First, we analyze the output conditioned on the adjacency matrix $\mathbf{A}$. Note that $\frac{1}{R}f_i^{(L)}(\mathbf{X})$ in Lemma A.9 is Lipschitz with constant $\sqrt{\frac{2}{\mathbf{deg}(i)}}$, and $h(\mathbf{X}_j)$ are mutually independent for $j\in[n]$. Therefore, by Gaussian concentration (Vershynin, 2018, Theorem 5.2.2) we have that for a fixed $i\in[n]$,

$$\mathbf{Pr}\left[\frac{1}{R}|f_i^{(L)}(\mathbf{X}) - \mathbb{E}[f_i^{(L)}(\mathbf{X})]| > \delta \mid \mathbf{A}\right] \le 2\exp\left(-\frac{\delta^2\mathbf{deg}(i)}{4\sigma^2}\right).$$

We refer to the event from Proposition A.1 as $B$ and define $Q(t)$ to be the event that

$$|f_i^{(L)}(\mathbf{X}) - \mathbb{E}[f_i^{(L)}(\mathbf{X})]| \le t \text{ for all } i\in[n].$$

Then we can write

$$\mathbf{Pr}\left[Q(t)^c\right] = \mathbf{Pr}\left[Q(t)^c\cap B\right] + \mathbf{Pr}\left[Q(t)^c\cap B^c\right]$$

$$\le 2n\exp\left(-\frac{t^2n(p+q)}{8\sigma^2}\right) + \mathbf{Pr}\left[B^c\right]$$

$$\le 2n\exp\left(-\frac{t^2n(p+q)}{8\sigma^2}\right) + 2n^{-c}.$$

Let $\xi = \text{sgn}(p-q)$ and note that $\frac{\xi(p-q)}{p+q} = \frac{|p-q|}{p+q} = \Gamma(p, q)$. We now choose $t = 2\sigma\sqrt{\frac{2(c+1)\log n}{n(p+q)}}$ to obtain that with probability at least $1 - 4n^{-c}$, the following holds for all $i \in [n]$:

$$
\begin{aligned}
\frac{1}{\sigma} f_i^{(L)}(\mathbf{X}) &= \mathbb{E}[f_i^{(L)}(\mathbf{X})]/\sigma \pm O\left(R\sqrt{\frac{c\log n}{n(p+q)}}\right) \\
&= \frac{R\xi}{\sigma \deg(i)} \sum_{j \in [n]} a_{ij}\mathbb{E}h(\mathbf{X}_j) \pm O\left(R\sqrt{\frac{c\log n}{n(p+q)}}\right) \\
&= \frac{\sqrt{2}R\xi\zeta(\gamma/2\sigma)}{\sigma \deg(i)} \left(\sum_{j \in C_1} a_{ij} - \sum_{j \in C_0} a_{ij}\right) \pm O\left(R\sqrt{\frac{c\log n}{n(p+q)}}\right) \quad \text{(Lemma A.4)} \\
&= \sqrt{2}(2\varepsilon_i - 1)R\Gamma(p, q)\zeta(\gamma/2\sigma)(1 \pm o_n(1)) \pm O\left(R\sqrt{\frac{c\log n}{n(p+q)}}\right) \quad \text{(Proposition A.1)} \\
&= \sqrt{2}(2\varepsilon_i - 1)R\Gamma(p, q)\zeta(\gamma/2\sigma)(1 \pm o_n(1)),
\end{aligned}
$$

where in the last equation we used the assumption that $\Gamma(p, q)\zeta(\gamma/2\sigma) = \omega\left(\sqrt{\frac{\log n}{n(p+q)}}\right)$. Overall, we obtain that with probability at least $1 - 4n^{-c}$,

$$
f_i^{(L)}(\mathbf{X}) = (2\varepsilon_i - 1)C\sigma R\Gamma(p, q)\zeta(\gamma/2\sigma)(1 \pm o_n(1)), \text{ for all } i \in [n].
$$

Recall that the loss for node $i$ is given by

$$
\ell_\theta^{(i)}(\mathbf{A}, \mathbf{X}) = \log(1 + e^{(1-2\varepsilon_i)f_i^{(L)}(\mathbf{X})}) = \log\left(1 + \exp\left(-C\sigma R\Gamma(p, q)\zeta(\gamma/2\sigma)(1 \pm o_n(1))\right)\right).
$$

The total loss is given by $\frac{1}{n}\sum_{i \in [n]} \ell_\theta^{(i)}(\mathbf{A}, \mathbf{X})$. Next, Fact A.5 implies that for any $t < 0$, $e^t/2 \leq \log(1 + e^t) \leq e^t$, hence, we have for some $C' \in [1/2, 1]$ that

$$
\ell_\theta(\mathbf{A}, \mathbf{X}) = C' \exp\left(-C\sigma R\Gamma(p, q)\zeta(\gamma/2\sigma)(1 \pm o_n(1))\right). \qquad \square
$$

### A.7 PROOF OF THEOREM 2 PART TWO

We begin by computing the output of the networks constructed in Proposition A.7 when two graph convolutions are placed among any layer in the networks other than the first.

**Lemma A.10.** *Let* $h(\boldsymbol{x}) : \mathbb{R}^d \to \mathbb{R} = |\langle \boldsymbol{x}, \hat{\boldsymbol{\nu}} \rangle| - |\langle \boldsymbol{x}, \hat{\boldsymbol{\mu}} \rangle|$. *Consider the networks constructed in Proposition A.7 equipped with two graph convolutions in the following combinations:*

1. *Both convolutions in the second layer of the two-layer network.*

2. *Both convolutions in the second layer of the three-layer network.*

3. *One convolution in the second layer and one in the third layer of the three-layer network.*

4. *Both convolutions in the third layer of the three-layer network.*

*Then for a sample* $(\mathbf{A}, \mathbf{X}) \sim \text{XOR-CSBM}(n, d, \boldsymbol{\mu}, \boldsymbol{\nu}, \sigma^2, p, q)$, *the output of the networks in all the above described combinations for a point* $i \in [n]$ *is*

$$
\hat{y}_i = \varphi(f_i^{(L)}(\mathbf{X})) = \varphi\left(\frac{R}{\deg(i)} \sum_{j \in [n]} \tau_{ij} h(\mathbf{X}_j)\right), \text{ where } \tau_{ij} = \sum_{k \in [n]} \frac{a_{ik}a_{jk}}{\deg(k)}.
$$

*Proof.* For the two-layer network, the output of the last layer when both convolutions are at the second layer is given by $f_i^{(2)}(\mathbf{X}) = (\mathbf{D}^{-1}\mathbf{A})^2[\mathbf{X}\mathbf{W}^{(1)}]_+\mathbf{W}^{(2)}$. Then we have

$$
f_i^{(2)}(\mathbf{X}) = \frac{R}{\deg(i)} \sum_{j \in [n]} \sum_{k \in [n]} \frac{a_{ij}a_{jk}}{\deg(j)} h(\mathbf{X}_k) = \frac{R}{\deg(i)} \sum_{j \in [n]} \tau_{ij} h(\mathbf{X}_j).
$$

Next, for the three-layer network, the output of the last layer when both convolutions are at the second layer is given by $f_i^{(3)}(\mathbf{X}) = [(\mathbf{D}^{-1}\mathbf{A})^2[\mathbf{X}\mathbf{W}^{(1)}]_+\mathbf{W}^{(2)}]_+\mathbf{W}^{(3)}$, hence, we have

$$
\begin{aligned}
f_i^{(3)}(\mathbf{X}) &= \frac{R}{\deg(i)} \left( \left[ \sum_{j \in [n]} \frac{a_{ij}}{\deg(j)} \sum_{k \in [n]} a_{jk} h(\mathbf{X}_k) \right]_+ - \left[ -\sum_{j \in [n]} \frac{a_{ij}}{\deg(j)} \sum_{k \in [n]} a_{jk} h(\mathbf{X}_k) \right]_+ \right) \\
&= \frac{R}{\deg(i)} \sum_{j \in [n]} \frac{a_{ij}}{\deg(j)} \sum_{k \in [n]} a_{jk} h(\mathbf{X}_k) \qquad (\text{using } [t]_+ - [-t]_+ = t \text{ for any } t \in \mathbb{R}) \\
&= \frac{R}{\deg(i)} \sum_{j \in [n]} \tau_{ij} h(\mathbf{X}_j).
\end{aligned}
$$

Similarly, the output of the last layer when one convolution is at the second layer and the other one is at the third layer is given by $f_i^{(3)}(\mathbf{X}) = \mathbf{D}^{-1}\mathbf{A}[\mathbf{D}^{-1}\mathbf{A}[\mathbf{X}\mathbf{W}^{(1)}]_+\mathbf{W}^{(2)}]_+\mathbf{W}^{(3)}$, hence, we have

$$
\begin{aligned}
f_i^{(3)}(\mathbf{X}) &= \frac{R}{\deg(i)} \sum_{j \in [n]} \frac{a_{ij}}{\deg(j)} \left( \left[ \sum_{k \in [n]} a_{jk} h(\mathbf{X}_k) \right]_+ - \left[ -\sum_{k \in [n]} a_{jk} h(\mathbf{X}_k) \right]_+ \right) \\
&= \frac{R}{\deg(i)} \sum_{j \in [n]} \frac{a_{ij}}{\deg(j)} \sum_{k \in [n]} a_{jk} h(\mathbf{X}_k) \qquad (\text{using } [t]_+ - [-t]_+ = t \text{ for any } t \in \mathbb{R}) \\
&= \frac{R}{\deg(i)} \sum_{j \in [n]} \tau_{ij} h(\mathbf{X}_j).
\end{aligned}
$$

Finally, the output of the last layer when both convolutions are at the third layer is given by $f_i^{(3)}(\mathbf{X}) = (\mathbf{D}^{-1}\mathbf{A})^2[[\mathbf{X}\mathbf{W}^{(1)}]_+\mathbf{W}^{(2)}]_+\mathbf{W}^{(3)}$, hence, we have

$$
\begin{aligned}
f_i^{(3)}(\mathbf{X}) &= \frac{R}{\deg(i)} \sum_{j \in [n]} \frac{a_{ij}}{\deg(j)} \left( \sum_{k \in [n]} a_{jk} \left( [h(\mathbf{X}_k)]_+ - [-h(\mathbf{X}_k)]_+ \right) \right) \\
&= \frac{R}{\deg(i)} \sum_{j \in [n]} \frac{a_{ij}}{\deg(j)} \sum_{k \in [n]} a_{jk} h(\mathbf{X}_k) = \frac{R}{\deg(i)} \sum_{j \in [n]} \tau_{ij} h(\mathbf{X}_j).
\end{aligned}
$$

Hence, the output for two graph convolutions is the same for any combination of the placement of convolutions, as long as no convolution is placed at the first layer. $\qquad \square$

We are now ready to prove the positive result for two convolutions.

**Theorem** (Restatement of part two of Theorem 2). *Let $(\mathbf{A}, \mathbf{X}) \sim$ XOR-CSBM$(n, d, \boldsymbol{\mu}, \boldsymbol{\nu}, \sigma^2, p, q)$. Assume that $p, q = \Omega(\frac{\log n}{\sqrt{n}})$ and $\Gamma(p,q)^2 \zeta(\gamma/2\sigma) = \omega\left(\sqrt{\frac{\log n}{n}}\right)$. Then for any $c > 0$, with probability at least $1 - O(n^{-c})$, the networks with any combination of two graph convolutions in the second and/or the third layers perfectly classify the data, and obtain the following loss:*

$$
\ell_\theta(\mathbf{A}, \mathbf{X}) = C' \exp\left( -C\sigma R\Gamma(p,q)^2 \zeta(\gamma/\sigma)(1 \pm \sqrt{c/\log n}) \right),
$$

*where $C > 0$ and $C' \in [1/2, 1]$ are constants and $R$ is the constraint from Eq. (1).*

*Proof.* The proof strategy is similar to that of part one of the theorem. Note that $\frac{1}{R} f_i^{(L)}(\mathbf{X})$ in Lemma A.10 is Lipschitz with constant

$$
\left\| \frac{1}{R} f_i^{(L)}(\mathbf{X}) \right\|_{\text{Lip}} \leq \sqrt{\frac{2}{\deg(i)^2} \sum_{j \in [n]} \tau_{ij}^2}.
$$

Since $h(\mathbf{X}_j)$ are mutually independent for $j \in [n]$, by Gaussian concentration (Vershynin, 2018, Theorem 5.2.2) we have that for a fixed $i \in [n]$,

$$\mathbf{Pr}\left[\frac{1}{R}|f_i^{(L)}(\mathbf{X}) - \mathbb{E}[f_i^{(L)}(\mathbf{X})]| > \delta \mid \mathbf{A}\right] \leq 2\exp\left(-\frac{\delta^2\mathbf{deg}(i)^2}{4\sigma^2\sum_{j\in[n]}\tau_{ij}^2}\right).$$

We refer to the event from Proposition A.2 as $B$. Note that since the graph density assumption is stronger than $\Omega(\frac{\log^2 n}{n})$, Proposition A.1 trivially holds in this regime, hence, the degrees also concentrate strongly around $\Delta = \frac{n}{2}(p+q)$. On event $B$, we have that

$$\sum_{j\in[n]}\tau_{ij}^2 = \sum_{j\in[n]}\left(\sum_{k\in[n]}\frac{a_{ik}a_{jk}}{\mathbf{deg}(k)}\right)^2 = \frac{1}{\Delta^2}\sum_{j\in[n]}\left(\sum_{k\in[n]}a_{ik}a_{jk}\right)^2 (1 \pm o_n(1))$$

$$= \frac{1}{\Delta^2}\left(\sum_{j\sim i}|N_i \cap N_j|^2 + \sum_{j\not\sim i}|N_i \cap N_j|^2\right)(1 \pm o_n(1))$$

$$= \frac{1}{\Delta^2}\left(\sum_{j\sim i}\left(\frac{n}{2}(p^2+q^2)\right)^2 + \sum_{j\not\sim i}(npq)^2\right)(1 \pm o_n(1)) \qquad \text{(using Proposition A.2)}$$

$$= \frac{n}{2\Delta^2}\left(\frac{n^2}{4}(p^2+q^2)^2 + n^2p^2q^2\right)(1 \pm o_n(1)) = \frac{n^3}{8\Delta^2}\left(p^4+q^4+6p^2q^2\right)(1 \pm o_n(1)).$$

Therefore, under this event we have that

$$\left\|\frac{1}{R}f_i^{(L)}(\mathbf{X})\right\|_{\mathrm{Lip}} \leq \sqrt{\frac{2}{\mathbf{deg}(i)^2}\sum_{j\in[n]}\tau_{ij}^2} = \sqrt{\frac{4(p^4+q^4+6p^2q^2)}{n(p+q)^4}}(1 \pm o_n(1)).$$

Note that $K = K(p,q) = \frac{4(p^4+q^4+6p^2q^2)}{(p+q)^4} \leq 4$. We now define $Q(t)$ to be the event that $|f_i^{(L)}(\mathbf{X}) - \mathbb{E}[f_i^{(L)}(\mathbf{X})]| \leq t$ for all $i \in [n]$. Then we have

$$\mathbf{Pr}\left[Q(t)^{\mathsf{c}}\right] = \mathbf{Pr}\left[Q(t)^{\mathsf{c}} \cap B\right] + \mathbf{Pr}\left[Q(t)^{\mathsf{c}} \cap B^{\mathsf{c}}\right] \leq 2n\exp\left(-\frac{nt^2}{2K^2\sigma^2}\right) + 2n^{-c}.$$

We now choose $t = \sigma\sqrt{\frac{2K(c+1)\log n}{n}}$ to obtain that with probability at least $1 - 4n^{-c}$, the following holds for all $i \in [n]$:

$$f_i^{(L)}(\mathbf{X}) = \mathbb{E}[f_i^{(L)}(\mathbf{X})] \pm O\left(R\sigma\sqrt{\frac{\log n}{n}}\right) = \frac{R}{\mathbf{deg}(i)}\sum_{j\in[n]}\tau_{ij}\mathbb{E}h(\mathbf{X}_j) \pm O\left(R\sigma\sqrt{\frac{\log n}{n}}\right).$$

Note that we have

$$\frac{1}{\mathbf{deg}(i)}\sum_{j\in[n]}\tau_{ij}\mathbb{E}h(\mathbf{X}_j) = \frac{\sqrt{2}\zeta(\gamma/2\sigma)}{\mathbf{deg}(i)}\left(\sum_{j\in C_1}\tau_{ij} - \sum_{j\in C_0}\tau_{ij}\right) \qquad \text{(using Lemma A.4)}$$

$$= \frac{\sqrt{2}\zeta(\gamma/2\sigma)}{\mathbf{deg}(i)}\left(\sum_{j\in C_1}\sum_{k\in[n]}\frac{a_{ik}a_{jk}}{\mathbf{deg}(k)} - \sum_{j\in C_0}\sum_{k\in[n]}\frac{a_{ik}a_{jk}}{\mathbf{deg}(k)}\right)$$

$$= \frac{\sqrt{2}\zeta(\gamma/2\sigma)}{\mathbf{deg}(i)}\left(\sum_{k\in[n]}\frac{a_{ik}}{\mathbf{deg}(k)}\left(\sum_{j\in C_1}a_{jk} - \sum_{j\in C_0}a_{jk}\right)\right)$$

$$= \frac{\sqrt{2}\zeta(\gamma/2\sigma)\Gamma(p,q)}{\mathbf{deg}(i)}\left(\sum_{k\in C_1}a_{ik} - \sum_{k\in C_0}a_{ik}\right)(1 + o_n(1))$$

$$= \sqrt{2}\zeta(\gamma/2\sigma)\Gamma(p,q)^2(1 + o_n(1)).$$

In the last two equations above, we used Proposition A.1 to replace, respectively,

$$\frac{1}{\mathbf{deg}(k)}\left(\sum_{j\in C_1} a_{kj} - \sum_{j\in C_0} a_{kj}\right) = (2\varepsilon_k - 1)\Gamma(p,q)(1+o_n(1)),$$

$$\frac{1}{\mathbf{deg}(i)}\left(\sum_{j\in C_1} a_{ik} - \sum_{j\in C_0} a_{ik}\right) = (2\varepsilon_k - 1)\Gamma(p,q)(1+o_n(1)).$$

Therefore, we obtain that

$$f_i^{(L)}(\mathbf{X}) = C\sigma R\zeta(\gamma/2\sigma)\Gamma(p,q)^2(1+o_n(1)) \pm O\left(R\sigma\sqrt{\frac{\log n}{n}}\right)$$

$$= C\sigma R\zeta(\gamma/2\sigma)\Gamma(p,q)^2(1+o_n(1)),$$

where in the last equation we used $\Gamma(p,q)^2\zeta(\gamma/2\sigma) = \omega\left(\sqrt{\frac{\log n}{n}}\right)$.

Recall that the loss for node $i$ is given by

$$\ell_\theta^{(i)}(\mathbf{A},\mathbf{X}) = \log(1 + \exp((1-2\varepsilon_i)f_i^{(L)}(\mathbf{X})))$$

$$= \log\left(1 + \exp\left(-C\sigma R\zeta(\gamma/2\sigma)\Gamma(p,q)^2(1\pm o_n(1))\right)\right).$$

The total loss is $\frac{1}{n}\sum_{i\in[n]}\ell_\theta^{(i)}(\mathbf{A},\mathbf{X})$. Now, using Fact A.5 we have for some $C' \in [1/2, 1]$ that

$$\ell_\theta(\mathbf{A},\mathbf{X}) = C'\exp\left(-C\sigma R\zeta(\gamma/2\sigma)\Gamma(p,q)^2(1\pm o_n(1))\right). \qquad \square$$

### A.8 ANALYSIS FOR A SIMPLER CASE

Although Theorem 2 encapsulates the general condition for networks with up to two graph convolutions to achieve perfect classification, let us discuss the meaning of the theorem in a simplified setting where $\Gamma(p,q) = \Omega(1)$. In this regime, one can analyze two cases for both parts of the theorem:

1. Case $\gamma = \Omega(\sigma)$: Using part two of Lemma A.4 implies that $\zeta(\gamma/\sigma) = \Omega(\frac{\gamma}{\sigma})$. Hence, for one graph convolution, the condition $\Gamma(p,q)\zeta(\gamma/2\sigma) = \omega\left(\sqrt{\frac{\log n}{n(p+q)}}\right)$ is satisfied when $\frac{\gamma}{\sigma} = \omega\left(\sqrt{\frac{\log n}{n(p+q)}}\right)$, implying that $\gamma = \omega\left(\sigma\sqrt{\frac{\log n}{n(p+q)}}\right)$. Similarly, for two graph convolutions, the condition $\Gamma(p,q)^2\zeta(\gamma/2\sigma) = \omega\left(\sqrt{\frac{\log n}{n}}\right)$ is satisfied when $\gamma = \omega\left(\sigma\sqrt{\frac{\log n}{n}}\right)$.

2. Case $\gamma = o(\sigma)$: Using part three of Lemma A.4 implies that $\zeta(\gamma/\sigma) = \Omega(\frac{\gamma^2}{\sigma^2})$. Hence, for one graph convolution, the condition $\Gamma(p,q)\zeta(\gamma/2\sigma) = \omega\left(\sqrt{\frac{\log n}{n(p+q)}}\right)$ is satisfied when $\left(\frac{\gamma}{\sigma}\right)^2 = \omega\left(\sqrt{\frac{\log n}{n(p+q)}}\right)$, implying that $\gamma = \omega\left(\sigma\sqrt[4]{\frac{\log n}{n(p+q)}}\right)$. Similarly, for two graph convolutions, the condition $\Gamma(p,q)^2\zeta(\gamma/2\sigma) = \omega\left(\sqrt{\frac{\log n}{n}}\right)$ is satisfied when $\gamma = \omega\left(\sigma\sqrt[4]{\frac{\log n}{n}}\right)$.

Combining both cases, we find that the theorems imply perfect classification if the following holds:

$$\gamma = \|\boldsymbol{\mu} - \boldsymbol{\nu}\|_2 = \begin{cases} \Omega\left(\frac{\sigma\sqrt{\log n}}{\sqrt[4]{n(p+q)}}\right) & \text{for networks wth one graph convolution,} \\ \Omega\left(\frac{\sigma\sqrt{\log n}}{\sqrt[4]{n}}\right) & \text{for networks with two graph convolutions.} \end{cases}$$

## B ADDITIONAL EXPERIMENTS

For all synthetic and real-data experiments, we used PyTorch Geometric (Fey & Lenssen, 2019), using public splits for the real datasets. The models were trained on an Nvidia Titan Xp GPU, using the Adam optimizer with learning rate $10^{-3}$, weight decay $10^{-5}$, and 50 to 500 epochs varying among the datasets.

## B.1 SYNTHETIC DATA

In this section we show additional results on the synthetic data. First, we show that placing a graph convolution in the first layer makes the classification task difficult since the means of the convolved data collapse towards 0. This is shown in Fig. 4.

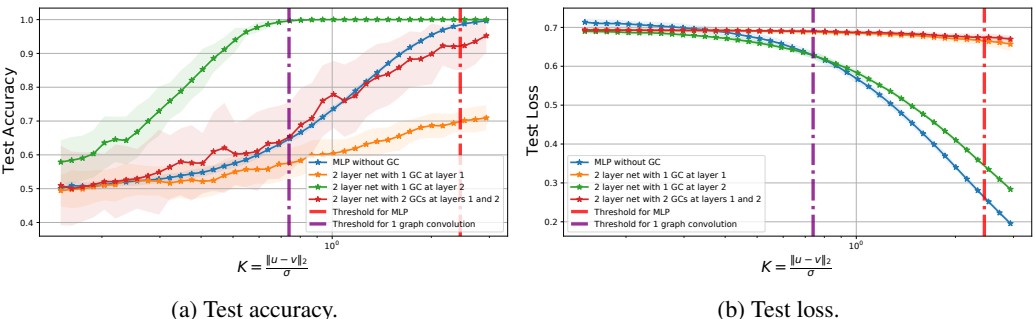

(a) Test accuracy.          (b) Test loss.

Figure 4: Comparing the accuracy and loss for various networks with and without graph convolutions, averaged over $50$ trials. Networks with a graph convolution in the first layer (red and orange) fail to generalize even for a large distance between the means of the data. For this experiment, we set $n = 400$ and $d = 4$, with $\sigma^2 = 1/d$.

Next, we show experiments for two sets of values of $p < q$ to demonstrate that graph convolutions also work in this setting. In Figs. 5a and 5b we have $\Gamma(p, q) \approx 0.82$, while in Figs. 5c and 5d we have a lower signal in the graph, $\Gamma(p, q) \approx 0.66$. We observe that in the latter case that there is less gap in the performance of networks with one graph convolution and those with two graph convolutions. In comparison to Fig. 2, we observe similar trends about the performance of all the networks in different regimes of interest. In particular, networks with one graph convolution perform mutually similarly, and networks with two graph convolutions perform mutually similarly, as claimed in Theorem 2.

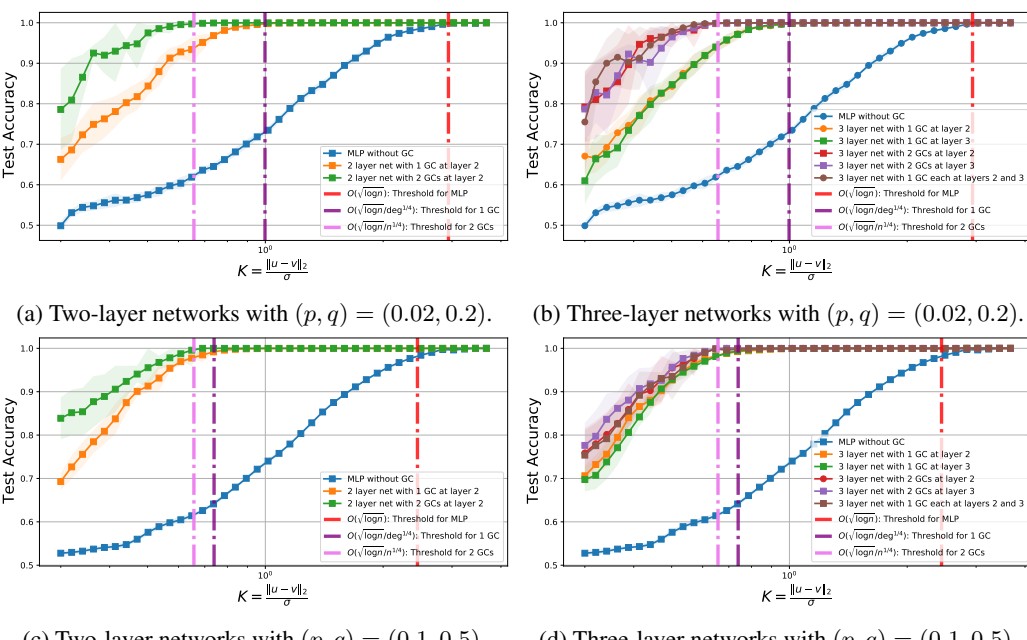

(a) Two-layer networks with $(p, q) = (0.02, 0.2)$.    (b) Three-layer networks with $(p, q) = (0.02, 0.2)$.

(c) Two-layer networks with $(p, q) = (0.1, 0.5)$.    (d) Three-layer networks with $(p, q) = (0.1, 0.5)$.

Figure 5: Averaged accuracy (over $50$ trials) for various networks with and without graph convolutions on the XOR-CSBM data model with $n = 400, d = 4$ and $\sigma^2 = 1/d$ for $p < q$. The x-axis denotes the ratio $K = \|\boldsymbol{\mu} - \boldsymbol{\nu}\|_2 / \sigma$ on a logarithmic scale. The vertical lines indicate the classification thresholds mentioned in part two of Theorem 1 (red), and in Theorem 2 (violet and pink).

Finally, in Fig. 6, we show the trends for the accuracy of various networks with and without graph convolutions, for different values of $\Gamma(p, q)$. For cases where $\Gamma(p, q)$ is relatively larger, networks with graph convolutions perform much better than a standard MLP (see Figs. 6a to 6d), while for the cases where $\Gamma(p, q)$ is much smaller, the networks with graph convolutions degrade in performance (see Figs. 6e to 6h). The intuition behind this behaviour is that a smaller value of $\Gamma(p, q)$ represents more noise in the data, thus, networks with graph convolutions gather roughly an equivalent amount of information from nodes in both the classes, making the feature representations noisy.

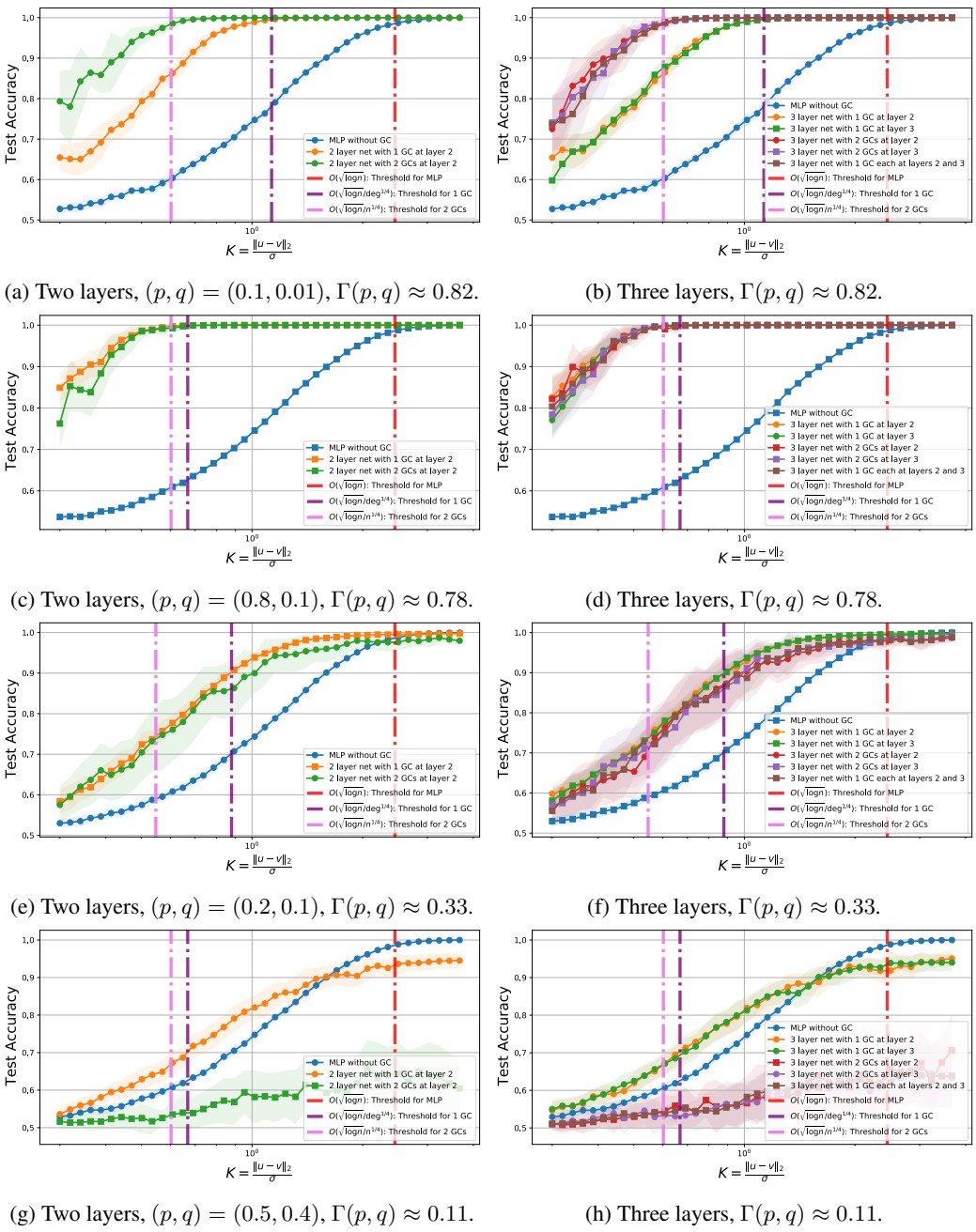

(a) Two layers, $(p, q) = (0.1, 0.01)$, $\Gamma(p, q) \approx 0.82$.

(b) Three layers, $\Gamma(p, q) \approx 0.82$.

(c) Two layers, $(p, q) = (0.8, 0.1)$, $\Gamma(p, q) \approx 0.78$.

(d) Three layers, $\Gamma(p, q) \approx 0.78$.

(e) Two layers, $(p, q) = (0.2, 0.1)$, $\Gamma(p, q) \approx 0.33$.

(f) Three layers, $\Gamma(p, q) \approx 0.33$.

(g) Two layers, $(p, q) = (0.5, 0.4)$, $\Gamma(p, q) \approx 0.11$.

(h) Three layers, $\Gamma(p, q) \approx 0.11$.

Figure 6: Test accuracy of various networks with with and without graph convolutions (GCs) for various values of $p$ and $q$, on the XOR-CSBM data model. Note that networks with graph convolutions degrade in performance as $\Gamma(p, q)$ (attributed to the signal in the graph) decreases.

## B.2 REAL-WORLD DATA

This section contains additional experiments on real-world data. In Fig. 7, we plot the accuracy of the networks measured on the three benchmark datasets, averaged across 50 different trials (random initialization of the network parameters). This corresponds to the plots in Fig. 3 that show the maximum accuracy across all trials. Next, we evaluate the performance of the original GCN

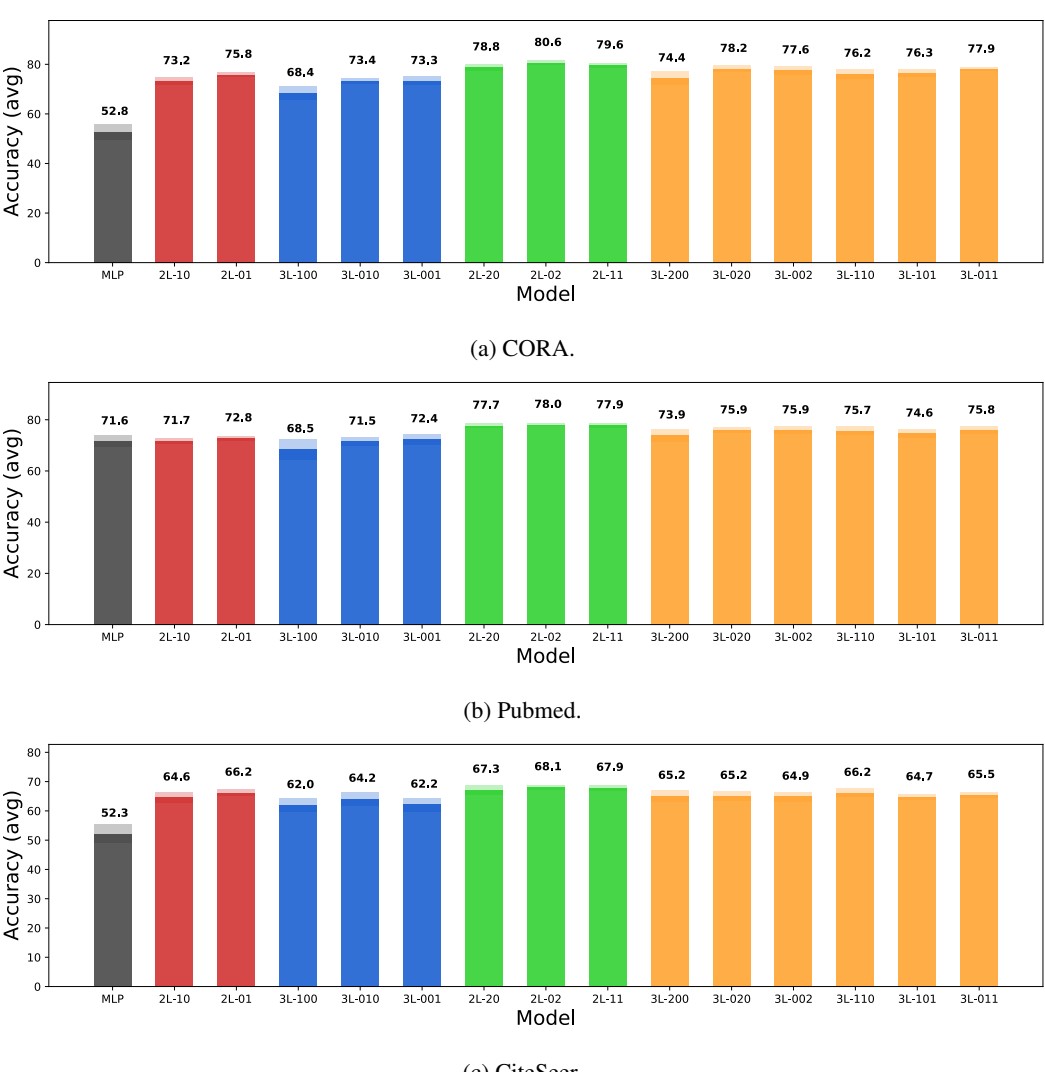

(a) CORA.

(b) Pubmed.

(c) CiteSeer.

Figure 7: Averaged accuracy (percentage) over 50 trials for various networks. A network with $k$ layers and $j_1, \ldots, j_k$ convolutions in each of the layers is represented by the label $k\text{L-}j_1 \ldots j_k$.

normalization (Kipf & Welling, 2017), $\mathbf{D}^{-\frac{1}{2}}\mathbf{A}\mathbf{D}^{-\frac{1}{2}}$ instead of $\mathbf{D}^{-1}\mathbf{A}$, and show that we observe the same trends about the number of convolutions and their placement. These results are shown in Figs. 8 and 9. Note the two general trends that are consistent: first, networks with two graph convolutions perform better than those with one graph convolution, and second, placing all graph convolutions in the first layer yields worse accuracy as compared to networks where the convolutions are placed in deeper layers.

Similar to the results in the main paper, we observe that there are differences within the group of networks with the same number of convolutions, however, these differences are smaller in magnitude as compared to the difference between the two groups of networks, one with one graph convolution and the other two graph convolutions. We also note that in some cases, three-layer networks obtain a

worse accuracy, which we attribute to the fact that three layers have a lot more parameters, and thus may either be overfitting, or may not be converging for the number of epochs used.

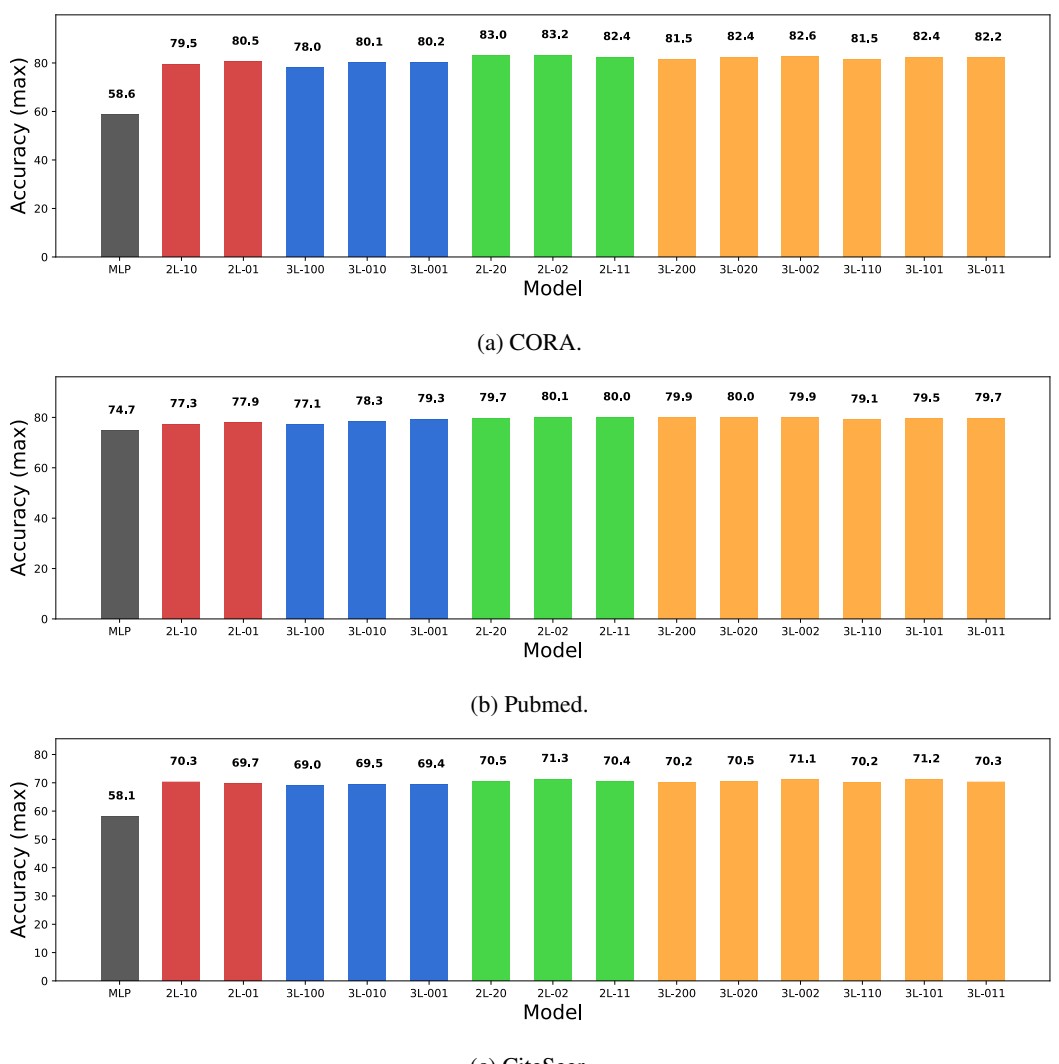

(a) CORA.

(b) Pubmed.

(c) CiteSeer.

Figure 8: Maximum accuracy (percentage) over 50 trials for various networks with the original GCN normalization $\mathbf{D}^{-\frac{1}{2}}\mathbf{A}\mathbf{D}^{-\frac{1}{2}}$. A network with $k$ layers and $j_1, \ldots, j_k$ convolutions in each of the layers is represented by the label $k\text{L-}j_1 \ldots j_k$.

Furthermore, we perform the same experiments on relatively larger datasets, OGBN-arXiv and OGBN-products (Hu et al., 2020). We observe similar trends in these experiments. First, we observe that networks with a graph convolution perform better than a simple MLP, and that two convolutions perform better than a single convolution. Furthermore, three graph convolutions do not have a significant advantage over two graph convolutions. This observation agrees with Lemma A.3, where one can compute $\rho(2)$ and $\rho(3)$ and realize that they are of the same order in $n$, i.e., the variance reduction offered by two graph convolutions is of the same order as three graph convolutions for sufficiently dense graphs. We present the results of these experiments in Fig. 10.

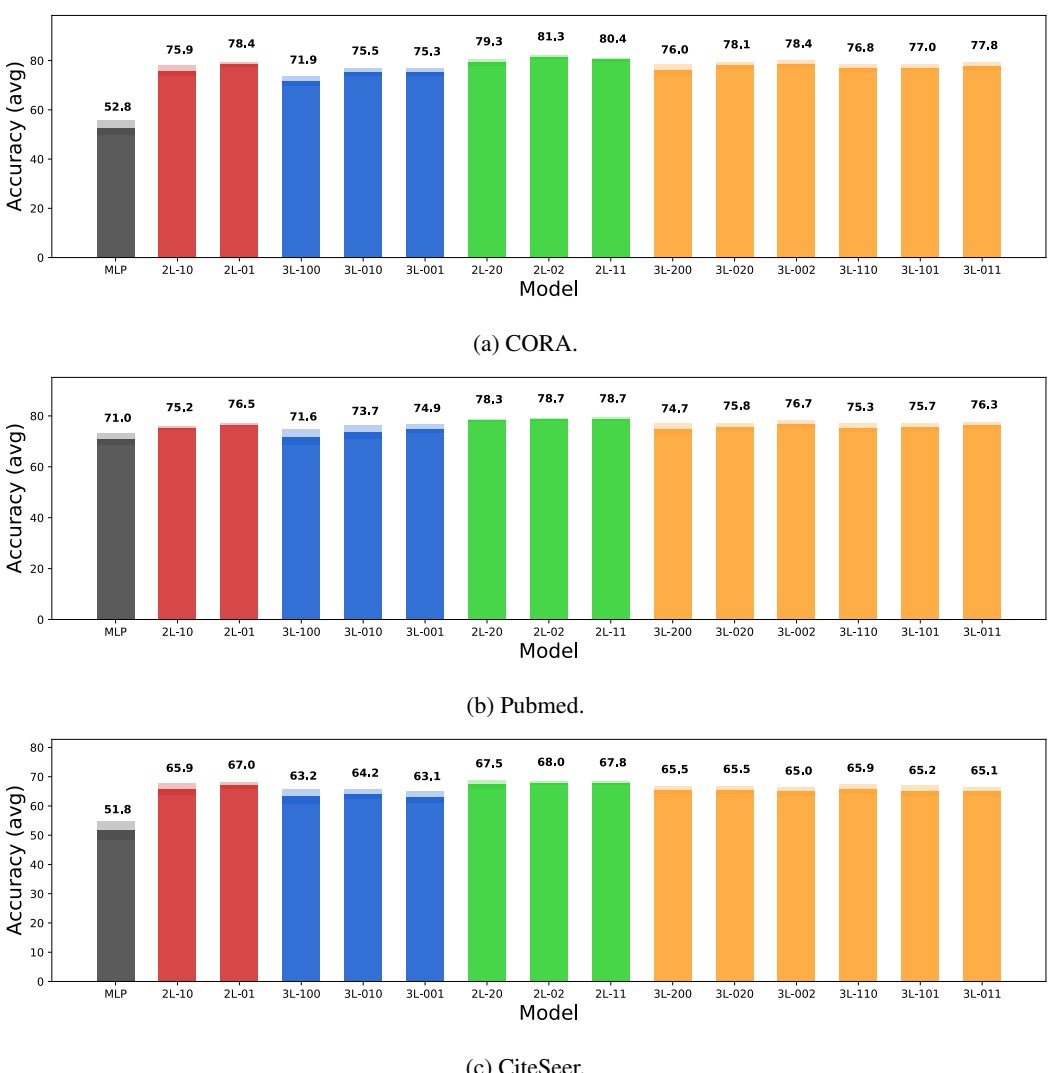

(a) CORA.

(b) Pubmed.

(c) CiteSeer.

Figure 9: Averaged accuracy (percentage) over 50 trials for various networks with the original GCN normalization $\mathbf{D}^{-\frac{1}{2}}\mathbf{A}\mathbf{D}^{-\frac{1}{2}}$. A network with $k$ layers and $j_1, \ldots, j_k$ convolutions in each of the layers is represented by the label $k\text{L-}j_1 \ldots j_k$.

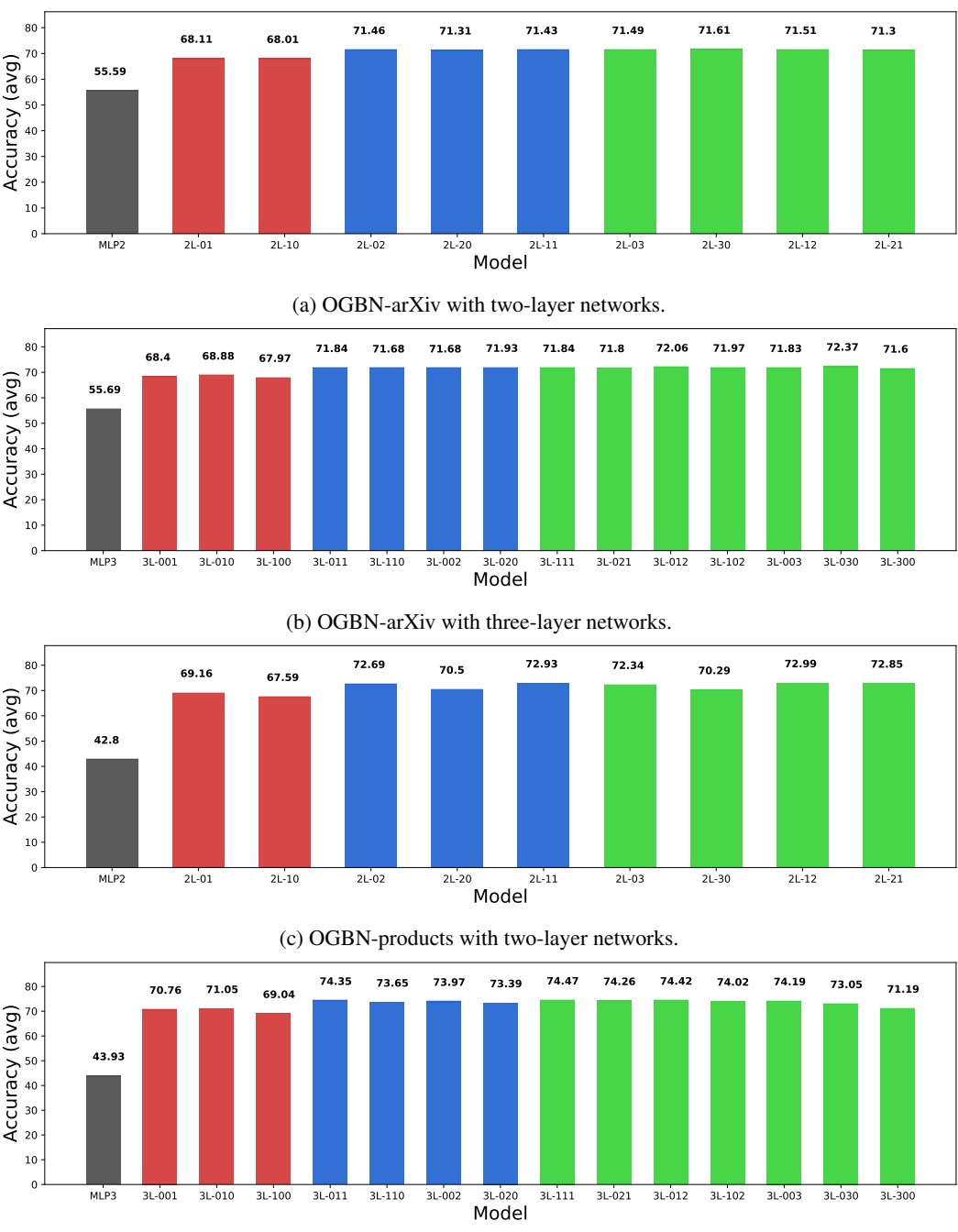

(a) OGBN-arXiv with two-layer networks.

(b) OGBN-arXiv with three-layer networks.

(c) OGBN-products with two-layer networks.

(d) OGBN-products with three-layer networks.

Figure 10: Averaged accuracy (percentage) for OGB datasets arXiv and products, over 10 trials for various networks. A network with $k$ layers and $j_1, \ldots, j_k$ convolutions in each of the layers is represented by the label $k$L-$j_1 \ldots j_k$, while MLP3 denotes a three-layer MLP. Note that all models with one GC (in red) perform mutually similarly, while models with two GCs (in blue) and three GCs (in green) perform mutually similarly and better than models with one GC.

