# OpenReview forum: "Effects of Graph Convolutions in Multi-layer Networks"
_ICLR.cc/2023/Conference — ICLR 2023 notable top 25%_

### Official Review · Reviewer_k2Qb · 2022-10-24

**Confidence:** 4
**Clarity, Quality, Novelty And Reproducibility:** The quality, clarity, and originality…
**Correctness:** 4
**Technical Novelty And Significance:** 4
**Empirical Novelty And Significance:** 4
**Recommendation:** 8

**Strength And Weaknesses:**

Strengths:
+ The motivation to study theoretically the function of graph convolution operators in multi-layer networks is interesting;
+ The theoretical findings as well as the corresponding analysis are beneficial for GNN researchers to get a better understanding on the effects of graph convolution layers;
+ The authors show that for multi-layer networks equipped with graph convolutions, the classification capacity is determined by the number of graph convolutions rather than the number of layers in the network. This is also interesting and can motivate more future work toward a more profound understanding of the capability of deep GNNs.

Weaknesses:
+ The assumption behind the theoretical analysis should be clarified. To what extent can the theoretical results be generalized to broader cases, instead of a non-linearly separable Gaussian mixture model coupled with a stochastic block model?
+ Potential limitation of the current theoretical analysis is not clear
+ As mentioned in the first point of contributions, a single graph convolution, and two graph convolutions both improve the threshold by a certain factor. What are the key differences between the theoretical analysis for these two cases? Is there any space to further expand the theoretical results to multiple graph convolutions? This may help interested readers have a clear mind on this line of research.

**Summary Of The Paper:**

This work provides a solid theoretical understanding of the role of graph convolutions (GC) in multi-layer neural nets. The theoretical analysis is based on the node classification problem of a non-linearly separable Gaussian mixture model coupled with a stochastic block model (SBM). Under some mild assumptions, the authors proved theoretically that, from various perspectives, GC is beneficial for improving the capability of multi-layer nets for the node classification task. Extensive experimental studies on both synthetic and real-world data are provided to validate the theoretical results.

**Summary Of The Review:**

Overall, I vote for accepting. The theoretical results and associated proofs are solid. The experimental results verify well their theories.

---

> ### Author Response · Authors · 2022-11-11
> **Response to questions**
>
> Dear reviewer, we are very grateful for your support and constructive feedback. We answer your questions below.
>
> > The assumption behind the theoretical analysis should be clarified. To what extent can the theoretical results be generalized to broader cases, instead of a non-linearly separable Gaussian mixture model coupled with a stochastic block model?
>
> All the assumptions related to our theoretical analysis are mentioned in the corresponding theorem statements and the definitions of the data model. The primary assumptions we make are about the nature of the data (GMM + SBM) and a lower bound on the order of $p,q$, that is, the sparsity of the graph. We expect that our results can be generalized to data models where instead of a two-class four-component mixture model, we have a general $l$-class $k$-component mixture model. In this setting, the threshold should be in terms of the minimum distance between the means across all pairs of the mixture. Our results stem from the fact that graph convolutions reduce the variance of the data and thus, reduce the overlap between different clusters of the mixture, and this fact applies to any $k$-component mixture in general.
>
> Furthermore, the results should be generalizable to distributions that are not strictly Gaussian, because all we require is a suitable form of concentration argument on the data distribution that helps us control the variance. We took the data to be Gaussian to simplify this part of the idea and bring out the main intuition about how graph convolutions affect the distribution. To study even more general distribution-agnostic models, there exist good theoretical tools like the PAC learning framework, however, they provide relatively weaker guarantees that may not be used to answer the specific questions that we study: (1) what are the benefits of two graph convolutions vs one graph convolution vs not using the graph, (2) does it matter where the graph convolutions are placed? For our assumptions on the data-generating distribution, we are able to answer these questions very precisely and are able to demonstrate that the results hold empirically on real data as well.
>
> > As mentioned in the first point of contributions, a single graph convolution, and two graph convolutions both improve the threshold by a certain factor. What are the key differences between the theoretical analysis for these two cases?
>
> The key difference between the theoretical analysis of one GC and two GCs is the quantity responsible for variance reduction. For one GC, the degree of each node signifies how strongly the output of the network concentrates around the expectation. Therefore, we require the degrees to concentrate for every node (Proposition A.1). For two GCs, the significant quantity is the number of common neighbours between a pair of nodes (Proposition A.2).
>
> > Is there any space to further expand the theoretical results to multiple graph convolutions? This may help interested readers have a clear mind on this line of research.
>
> Thank you for this question! We agree that an extension to multiple graph convolutions will help interested readers in this line of research. The first step for analyzing $K>2$ convolutions in our framework is to understand the variance reduction obtained by the $K$ convolutions. We present this first step in Lemma A.3 and the paragraph that follows, where the variance reduction is characterized as $\rho(K)$. However, $\rho(K)$ is in terms of $A^K(i,j)$, the entries of the exponentiated adjacency matrix. These values are highly correlated for larger values of $K$, and hence, obtaining high probability guarantees for $\rho(K)$ requires more advanced tools on the concentration of measure phenomena. Furthermore, for larger values of $K$, it is important to understand the tradeoff between two effects of GCs: (1) reduction in the distance between the means, and (2) reduction in the variance. For $K=1,2$, this tradeoff is simple enough to compute and helps us obtain the claimed results, while for larger $K$, we have the problem of the random variables being highly correlated. Thus, we leave the complete study of the effects of more convolutions as a potentially interesting future work.

---

### Official Review · Reviewer_4qct · 2022-10-25

**Confidence:** 3
**Correctness:** 3
**Technical Novelty And Significance:** 3
**Empirical Novelty And Significance:** 3
**Recommendation:** 8

**Clarity, Quality, Novelty And Reproducibility:**

- The paper is well-written and easy to follow. All notations are clearly defined.

- The idea to theoretically investigate the superiority of GNNs compared with MLPs and the combination of graph convolutions seems novel and promising to me.

- The experiments seem easy to reproduce.

**Strength And Weaknesses:**

### Strength

- The paper is well-written and easy to follow. The mathematical notations are clarified.
- The paper theoretically shows that GNNs are better than MLPs for graph data under some preconditions (even though it may be a little strong).
- The provided insight about "what really matters is the number of graph operations rather than the way to combine it" is interesting and impressive.

### Weakness

- The fundamental assumption of $X$ is that each feature vector is sampled from the XOR mixture of Gaussian models, XOR-GMM. Maybe it is strong for most cases. In particular, the graph is usually used for non-Euclidean data. What's more, why do we need the random variable $\eta$ in XOR-GMM?
- As the discussion is limited to XOR-GMM (Part-1 of Theorem 1), the authors only discuss the two-layer and three-layer cases, which is a little tricky.
- It might be inadequate to use the same bound of both $p$ and $q$, since there should be a difference between the inter-class and intra-class probability.
- The theoretical conclusion concentrates on the binary classification. Could the theory be generalized into the multi-class case?
- In the footnotes of Page-3, the authors claimed that " they readily generalize to other normalization methods". Could the authors provide more discussions about how to generalize the shown proof to other normalization methods? I'm curious about this question but I could not check the proof very carefully due to the tight review deadline.
- The readability could be further improved. For example,
    - The definition of $\gamma$ could be introduced in Theorem 1, so that the readers could find the variation of $\gamma$ in both Theorem 1 and Theorem 2.
    - At the end of the 3rd paragraph in Section 2.1, the subscript of $\{X_i\}$ should be $i\in[n]$.

**Summary Of The Paper:**

This paper provides some theoretical conclusions, including:
- GNNs are better than MLPs on graph data provided that node features are sampled from XOR-GNN (though it may be a little strong).
- Any combinations of graph convolutions have similar performance as long as the number/order of graph convolutions is the same.
Some experiments are also conducted to verify the theoretical conclusions.

**Summary Of The Review:**

I would like to update my score after reading other reviews and the response from the authors.

---

> ### Author Response · Authors · 2022-11-11
> **Response to questions - part 2**
>
> > In the footnotes of Page-3, the authors claimed that "they readily generalize to other normalization methods". Could the authors provide more discussions about how to generalize the shown proof to other normalization methods?
>
> Due to degree concentration, i.e., every node's degree is roughly the same under our assumptions on $p,q$ (Proposition A.1), any degree-based normalization obtains the same effect. We analyze the convolution $D^{-1}AX$, which implies that the transformed representation is $\tilde{X}\_i = \frac{1}{D\_{ii}}\sum_{j\in[n]}A\_{ij}X\_j$ for input representations $\\{X\_i\\}\_{i\in[n]}$. Another popular normalization is $D^{-1/2}AD^{-1/2}X$, which implies that the transformed representation is $\tilde{X}\_i = \frac{1}{\sqrt{D\_{ii}}}\sum\_{j\in[n]}\frac{1}{\sqrt{D\_{jj}}}A\_{ij}X\_j$. The proof can be easily adapted to this normalization because all $D_{ii}$ concentrate strongly around the expected degree. We also empirically demonstrate in Section B.2 that the observations are similar for the two kinds of normalizations.
>
> > The readability could be further improved.
>
> Thank you for pointing out the typo and suggesting readability improvements. We agree with you and have fixed the typo, along with the suggested addition of $\gamma$ in theorem 1.

---

> ### Author Response · Authors · 2022-11-11
> **Response to questions - part 1**
>
> Dear reviewer, we sincerely thank you for your encouraging words and valuable, constructive feedback. We answer the questions below. Please let us know if you have any follow-up questions.
>
> > The fundamental assumption of $X$ is that each feature vector is sampled from the XOR mixture of Gaussian models, XOR-GMM. Maybe it is strong for most cases.
>
> We chose the XOR-GMM as the toy model to set up the baseline for our theoretical results because it is the simplest model that requires multiple layers, however, we agree with you that the assumption is strong for practical cases. We expect that our results can be generalized to data models where instead of a two-class four-component mixture model, we have a general $l$-class $k$-component mixture model. In this setting, the threshold should be in terms of the minimum distance between the means across all pairs of the mixture. Our results stem from the fact that graph convolutions reduce the variance of the data and thus, reduce the overlap between different clusters of the mixture, and this fact applies to any $k$-component mixture in general.
>
> Furthermore, the results should be generalizable to distributions that are not strictly Gaussian, because all we require is a suitable form of concentration argument on the data distribution that helps us control the variance. We took the data to be Gaussian to simplify this part of the idea and bring out the main intuition about how graph convolutions affect the distribution. Please also see that we do not restrict our experiments to synthetic data. We test the methods extensively on well-known benchmark datasets for multi-class node classification.
>
> > Why do we need the random variable $\eta$ in XOR-GMM?
>
> There are four clusters in the XOR-GMM. Two of these clusters with means $\pm\mu$ are in $C_0$ and the two with means $\pm\nu$ are in $C_1$. The variable $\eta_i$ determines whether the point $i$ is from a Gaussian with mean in $\\{\mu,\nu\\}$ or $\\{-\mu,-\nu\\}$.
>
> > As the discussion is limited to XOR-GMM (Part-1 of Theorem 1), the authors only discuss the two-layer and three-layer cases, which is a little tricky.
>
> The data models that we study can be classified with two-layer and three-layer networks. We agree with you that practically, this limitation might be substantial. However, please note that although our theoretical discussion is constrained to XOR-CSBM, our empirical study spans a variety of benchmark datasets for multi-class node classification. The results obtained on these datasets align with the theoretical insights we present in our theorems. Furthermore, for these benchmark datasets, the state-of-the-art GCN implementation requires up to 3 layers (Semi-Supervised Classification with Graph Convolutional Networks, Kipf and Welling, ICLR 2017).
>
> As answered previously, generalization of the idea should be possible for an $l$-class $k$-component mixture model. To study even more general distribution-agnostic models, there exist good theoretical tools like the PAC learning framework, however, they provide relatively weaker guarantees that may not be used to answer the specific questions that we study: (1) what are the benefits of two graph convolutions vs one graph convolution vs not using the graph, (2) does it matter where the graph convolutions are placed? For our assumptions on the data-generating distribution, we are able to answer these questions very precisely and are able to demonstrate that the results hold empirically on real data as well.
>
> > It might be inadequate to use the same bound of both $p$ and $q$, since there should be a difference between the inter-class and intra-class probability.
>
> We agree that there should be a difference between the intra-class and inter-class probability. In the statement of theorem 2, this difference is captured by the signal $\Gamma(p,q)=\frac{|p-q|}{p+q}$. The lower bound for both $p$ and $q$, i.e., $\Omega(\log^2n/n)$ is asymptotic and exists only to guarantee degree concentration. As such, this lower bound is not related to the difference between $p$ and $q$. The difference is captured by the quantity $\Gamma(p,q)$.
>
> > The theoretical conclusion concentrates on the binary classification. Could the theory be generalized into the multi-class case?
>
> We expect that our results can be generalized to data models where instead of a two-class four-component mixture model, we have a general $l$-class $k$-component mixture model. In this setting, the threshold should be in terms of the minimum distance between the means across all pairs of the mixture. Our results stem from the fact that graph convolutions reduce the variance of the data and thus, reduce the overlap between different clusters of the mixture, and this fact applies to any $k$-component mixture in general. We also demonstrate insights from our theoretical results extensively on well-known benchmark datasets for multi-class node classification.

---

### Official Review · Reviewer_vTDa · 2022-10-25

**Confidence:** 4
**Correctness:** 4
**Technical Novelty And Significance:** 3
**Empirical Novelty And Significance:** 2
**Recommendation:** 8

**Clarity, Quality, Novelty And Reproducibility:**

The paper is clearly written. The theoretical arguments and observations are backed by good quality discussions. I did not check the proofs rigorously, however, the theoretical findings on SBM design for the node-classification task are along expected lines. Insights into the placement of graph convolutions in various layers of the network are illuminating.

**Strength And Weaknesses:**

Strengths:

This paper enhances the understanding of graph convolutions in learnability. Although the considered node classification problem is parametric and simpler than what is usually encountered in practice, the results in this paper are backed by rigorous theoretical analyses. Experimental validations on synthetic and real-world datasets are also provided.

Weaknesses:
I did not identify any major weaknesses. However, I could not understand how the hyper-parameters in the experiments were selected.

**Summary Of The Paper:**

This paper studies the impact of graph convolutions on a binary node classification problem. In this setting, the relationships among nodes are sampled from a given stochastic block model and node features are associated with a Gaussian mixture model. Theoretical results on the performance of a two or three layer graph convolutional network on the node classification task are provided. Additionally, the performance is characterized in terms of number of graph convolutions and their placement. Interestingly, it is shown that placement of graph convolution in the first layer jeopardizes the node classification performance. Theoretical results are validated via experiments on synthetic and real world datasets.

**Summary Of The Review:**

This paper provides a solid theoretical contribution on the learnability of graph convolutional networks for a node classification task. My initial recommendation is accept.

---

> ### Author Response · Authors · 2022-11-11
> **Response to question about hyperparameters**
>
> Dear reviewer, we sincerely thank you for your encouraging words and are very grateful for your support. We explain our choice of hyperparameters below.
>
> > However, I could not understand how the hyper-parameters in the experiments were selected.
>
> The selected hyperparameters are directly taken from state-of-the-art implementations of vanilla GCN used in the original GCN paper (Kipf and Welling, 2017) for the small citation networks CORA, CiteSeer and Pubmed. For the OGB datasets, we tried several sets of hyperparameters and chose the one that gave the best test error averaged over 10 trials. The actual values of the hyperparameters are mentioned within the description of each experiment in Section 4 and Section B.

---

### Official Review · Reviewer_Wv5W · 2022-10-26

**Confidence:** 5
**Correctness:** 3
**Technical Novelty And Significance:** 3
**Empirical Novelty And Significance:** 2
**Recommendation:** 6

**Clarity, Quality, Novelty And Reproducibility:**

Clarity: good
Quality: good
Novelty: medium
Reproducibility: good

**Strength And Weaknesses:**

## Strength
The analysis is comprehensive.

## Weaknesses
1. The assumption of the analysis is over-simplified.
2. Some claims and experimental results are only valid for homophilic graphs, but not for heterophilic graphs. But I think it has the potential to be extended to homophily/heterophily problem.

## Questions and Comments

1. The 4th line in the first equation in section 2.2, is that f^L or H^L?

2. How is your assumption over p,q related to homophily/heterophily problem identified in [1,2], e.g. heterophily is not always harmful and homophily is not always necessary for GNNs.

3. How is your analysis generalized to more complex settings, e.g. multi-class classification or more general conditions on data generation distributions.

4. “the placement of graph convolutions does not matter as long as it is not in the first layer”. Why the graph convolutions cannot be put into the first layer? How does this align with your previous analysis?

5. “as claimed in Theorem 2, networks that utilize the graph perform remarkably better than a traditional MLP that does not use relational information.” I don’t think this conclusion is still valid on heterophilic graphs, see [2].

6. Why not do more convolutions, e.g. 5 or 10? Do your claim and analysis still hold?

[1] Ma Y, Liu X, Shah N, et al. Is homophily a necessity for graph neural networks?[J]. arXiv preprint arXiv:2106.06134, 2021.

[2] Luan S, Hua C, Lu Q, et al. Is Heterophily A Real Nightmare For Graph Neural Networks To Do Node Classification?[J]. arXiv preprint arXiv:2109.05641, 2021.



**Summary Of The Paper:**

The authors try to prove that the performance of graph convolutions placed in different combinations among the layers of a neural network is mutually similar for all combinations of the placement.

**Summary Of The Review:**

The theoretical analysis looks fancy, but some claims and conclusions seem trivial. The significance of this paper is not as strong as its analysis. It is a borderline paper.

---

> ### Public Comment · ~Pan_Li2 · 2022-11-05
> **One note on heterophilic networks**
>
> Thank the authors for having this interesting paper and also thank the reviewer for the insightful comments on homophilic and heterophilic graphs. I would like to highlight one work [1] that unifies homophilic and heterophilic graphs, while only on one layer GNN.
>
> [1] Understanding Non-linearity in Graph Neural Networks from the Bayesian-Inference Perspective, NeurIPS 2022, Rongzhe Wei, Haoteng Yin, Junteng Jia, Austin R. Benson, Pan Li

---

> > ### Author Response · Authors · 2022-11-11
> > **Thank you for highlighting your work**
> >
> > Thank you Prof. Li for highlighting your interesting work. This is quite related to our study and we have added a citation to your paper. It is interesting to see that your results also apply to both homophilous and heterophilous settings. You also consider the SNR for the Gaussian mixture to be the ratio of the squared distance between the means to the variance. Furthermore, you identify that the gap $|p-q|$ reflects structural information, which aligns with what we identify as the signal in the graph, i.e., $\Gamma(p,q)=\frac{|p-q|}{p+q}$. We also thank you for acknowledging that our results go beyond one layer.

---

> ### Author Response · Authors · 2022-11-11
> **Response to questions - part 2**
>
> > "the placement of graph convolutions does not matter as long as it is not in the first layer". Why the graph convolutions cannot be put into the first layer? How does this align with your previous analysis?
>
> We chose the XOR model to demonstrate an important phenomenon that graph convolutions in the first layer perform an averaging on the original node representations, which can lead to a loss in class-membership information. On the other hand, if we have graph convolutions in later layers, the averaging happens on transformed node representations, which preserves the class-membership information and obtains the desired variance reduction. For the XOR model, the features of the two classes will collapse to the same point upon convolution. Hence, in the first layer, a GC averages the original features of the neighbours, making the transformed distributions for both classes indistinguishable with high probability. Please also refer to the final paragraph of Section 3.2 and Figure 1 where we explain this intuition. For a formal argument, please refer to Section A.5. We also demonstrate this phenomenon empirically in Section B.1.
>
> > "as claimed in Theorem 2, networks that utilize the graph perform remarkably better than a traditional MLP that does not use relational information." I don’t think this conclusion is still valid on heterophilic graphs, see [2].
>
> > Some claims and experimental results are only valid for homophilic graphs, but not for heterophilic graphs. But I think it has the potential to be extended to homophily/heterophily problem.
>
> We would like to clarify that the conclusion of theorem 2 is valid for both homophilic and heterophilic graphs. The theorem states that for our XOR-CSBM data model, under the given sparsity assumptions, networks with one and two GCs perform better than a traditional MLP in terms of the threshold for perfect classification. We prove this result formally in Sections A.6 and A.7 for both cases: $p > q$ (homophilic graph) and $p < q$ (heterophilic graph). However, we agree that there may exist datasets where heterophily is a problem (e.g. Cornell, Texas, Pei et al. 2020), as described in [1,2]. But please note that we do not claim anything about heterophilic graphs in general, as this is not the focus of our study. Our statements and empirical observations are valid for the datasets that we work with, and our theorems are valid for the data model that we study.
>
> > Why not do more convolutions, e.g. 5 or 10? Do your claim and analysis still hold?
>
> The first step to analyze $K>2$ convolutions in our framework is to understand the variance reduction obtained by the $K$ convolutions. We present this first step in Lemma A.3 and the paragraph that follows, where the variance reduction is characterized as $\rho(K)$. However, $\rho(K)$ is in terms of $A^K(i,j)$, the entries of the exponentiated adjacency matrix. These values are highly correlated for larger values of $K$, and hence, obtaining high probability guarantees for $\rho(K)$ requires more advanced tools on the concentration of measure phenomena. Furthermore, for larger values of $K$, it is important to understand the tradeoff between two effects of GCs: (1) reduction in the distance between the means, and (2) reduction in the variance. For $K \le 2$ this tradeoff is simple enough to compute and helps us obtain the claimed results, while for larger $K$, we have the problem of the random variables being highly correlated. Thus, we leave the complete study of the effects of more convolutions as potential future work.

---

> ### Author Response · Authors · 2022-11-11
> **Response to questions - part 1**
>
> Dear reviewer, we thank you for your valuable feedback and constructive questions. We answer the concerns below.
>
> > The 4th line in the first equation in section 2.2, is that f^L or H^L?
>
> The 4th line is $\hat{\bf y} = \varphi(f^{(L)})$. In the actual implementation, $H^{(L)}$ is not computed. We only compute up to $H^{(L-1)}$ and in the last layer, we apply sigmoid activation instead of ReLU.
>
> > How is your assumption over p,q related to homophily/heterophily problem identified in [1,2], e.g. heterophily is not always harmful and homophily is not always necessary for GNNs.
>
> Our assumptions on $p,q$, i.e., $p,q=\Omega(\log^2n/n)$ are present to only lower bound the expected degree and are not related to homophily/heterophily. Regarding the homophily/heterophily settings in [1,2], please note that our results hold for both of these cases. In particular, in [1,2], homophily ratio is defined to be the fraction of edges that connect nodes with the same label. In our setting, this ratio is $\frac{p}{p+q}$. A graph is said to be homophilic if the homophily ratio $>0.5$, and heterophilic if it is $<0.5$. For our setting this translates to $p > q$ (homophilic graph) and $p < q$ (heterophilic graph), where $p$ and $q$ are the intra-class and inter-class edge probabilities. Our theoretical results are valid for both of these cases. This is why in the theorem statements, the relevant SNR $\Gamma(p,q)=\frac{|p-q|}{p+q}$ is always positive irrespective of the sign of $p-q$.
>
> We agree with you on the fact that heterophily is not always harmful and homophily is not always necessary for GNNs. In fact, our results serve as one such example where heterophily is not harmful and homophily is not necessary. The intuition behind why our results hold for both cases is that for our setting, a network with a graph convolution finds an optimal classifier that is proportional to ${\rm sgn}(p-q)$ (please see Lemma A.9). Hence, the architecture works the same way and obtains the same variance reduction for both $p > q$ and $p < q$. Graph convolution is an averaging operation, and to compute the feature representation of a node in, say, class $C_0$, it gathers more information from nodes in $C_0$ than in $C_1$ for $p > q$, and more information from nodes in $C_1$ than in $C_0$ for $p < q$. In both cases, it performs a variance reduction on the data which is the key to improved performance. The work highlighted in the comment by Prof. Pan Li is another example where the results apply to both homophilic and heterophilic settings, although for one-layer networks.
>
> We also note that heterophily might be an issue in some settings and observations, as demonstrated in [1,2], however, in our setting where we work with the XOR-CSBM data model, the results hold for both kinds of graphs. We have added the relevant citations and a brief discussion on this subject in the related work section.
>
> > How is your analysis generalized to more complex settings, e.g. multi-class classification or more general conditions on data generation distributions.
>
> We expect that our results can be generalized to data models where instead of a two-class four-component mixture model, we have a general $l$-class $k$-component mixture model. In this setting, the threshold should be in terms of the minimum distance between the means across all pairs of the mixture. Our results stem from the fact that graph convolutions reduce the variance of the data and thus, reduce the overlap between different clusters of the mixture, and this fact applies to any $k$-component mixture in general.
>
> Furthermore, the results should be generalizable to distributions that are not strictly Gaussian, because all we require is a suitable form of concentration argument on the data distribution that helps us control the variance. We took the data to be Gaussian to simplify this part of the idea and bring out the main intuition about how graph convolutions affect the distribution. Please also see that we do not restrict our experiments to 2 classes or to synthetic data. We test the methods extensively on well-known benchmark datasets for multi-class node classification.
>
> To study even more general distribution-agnostic models, there exist good theoretical tools like the PAC learning framework, however, they provide relatively weaker guarantees that may not be used to answer the specific questions that we study: (1) what are the benefits of two graph convolutions vs one graph convolution vs not using the graph, (2) does it matter where the graph convolutions are placed? For our assumptions on the data-generating distribution, we are able to answer these questions very precisely and are able to demonstrate that the results hold empirically on real data as well.

---

### Decision · Program_Chairs · 2023-01-20

**Decision:**

Accept: notable-top-25%

**Justification For Why Not Higher Score:**

NA

**Justification For Why Not Lower Score:**

All reviewers appreciate the theoretical contributions of this work, and some concerns have been clarified during rebuttal. Overall, the work is very solid and is a significant contribution to the machine learning community.

**Metareview: Summary, Strengths And Weaknesses:**

This paper provides a theoretical study of the effects of graph convolution in deep nets. All reviewers appreciate the theoretical contributions of this work, and some concerns have been clarified during rebuttal. Overall, the work is very solid and is a significant contribution to the machine learning community.

**Note From Pc:**

if the above contains the word "oral" or "spotlight" please see: "oral" presentation means -> notable-top-5% and "spotlight" means -> notable-top-25%. As stated in our emails, we are disassociating presentation type from AC recommendations